# Hydrogels with tunable mechanical plasticity regulate endothelial cell outgrowth in vasculogenesis and angiogenesis

Zhao Wei[1,2,4], Meng Lei[1,2,4], Yaohui Wang[1,2], Yizhou Xie[1,2], Xueyong Xie[1,2], Dongwei Lan[1,2], Yuanbo Jia[1,2], Jingyi Liu[1,2], Yufei Ma [1,2], Bo Cheng[1,2], Sharon Gerecht [3] ✉ & Feng Xu [1,2] ✉

The endothelial cell (EC) outgrowth in both vasculogenesis and angiogenesis starts with remodeling surrounding matrix and proceeds with the crosstalk between cells for the multicellular vasculature formation. The mechanical plasticity of matrix, defined as the ability to permanently deform by external traction, is pivotal in modulating cell behaviors. Nevertheless, the implications of matrix plasticity on cell-to-cell interactions during EC outgrowth, along with the molecular pathways involved, remain elusive. Here we develop a collagen-hyaluronic acid based hydrogel platform with tunable plasticity by using compositing strategy of dynamic and covalent networks. We show that although the increasing plasticity of the hydrogel facilitates the matrix remodeling by ECs, the largest tubular lumens and the longest invading distance unexpectedly appear in hydrogels with medium plasticity instead of the highest ones. We unravel that the high plasticity of the hydrogels promotes stable integrin cluster of ECs and recruitment of focal adhesion kinase with an overenhanced contractility which downregulates the vascular endothelial cadherin expression and destabilizes the adherens junctions between individual ECs. Our results, further validated with mathematical simulations and in vivo angiogenic tests, demonstrate that a balance of matrix plasticity facilitates both cell-matrix binding and cell-to-cell adherens, for promoting vascular assembly and invasion.

Many vascular diseases and disorders, ranging from mild Raynaud's disease or chronic wounds to severe peripheral artery disease and even mortally ischemic stroke, still bring suffering worldwide[1,2]. Vasculature formation is the morphogenesis of endothelial cells (ECs) into luminal outgrowth around the lesion through either coalition of individual ECs (i.e., vasculogenesis) or the sprouting of ECs from pre-existing vessels (i.e., angiogenesis). EC outgrowth in both vasculogenesis and angiogenesis has shown great potential for treating vessel-related diseases[3,4]. The ECs in vivo are surrounded by extracellular matrix (ECM), and the crosstalk between ECs and ECM plays a vital role in regulating EC outgrowth[5,6]. Therefore, it is important to uncover the effects of ECM cues on EC outgrowth, as well as the underlying mechanisms, for desirably governing vasculature formation[7].

Mechanosensing of ECM has become a key factor influencing EC behaviors, leading to the development of various engineered elastic hydrogels with tailored mechanical properties for EC outgrowth[8–13].

[1]The Key Laboratory of Biomedical Information Engineering of Ministry of Education, School of Life Science and Technology, Xi'an Jiaotong University, Xi'an 710049, P.R. China. [2]Bioinspired Engineering and Biomechanics Center (BEBC), Xi'an Jiaotong University, Xi'an 710049, P.R. China. [3]Department of Biomedical Engineering, Duke University, Durham, NC 27708, USA. [4]These authors contributed equally: Zhao Wei, Meng Lei. ✉e-mail: sharon.gerecht@duke.edu; fengxu@mail.xjtu.edu.cn

For example, the progression of EC morphogenesis occurs in a synthetic hyaluronic acid (HA) hydrogel both in vitro and in vivo with a selected matrix stiffness of around 200 Pa[14], while a stiffer matrix (gelatin methacryloyl hydrogel) boosts ECs sprouting from a spheroid model[15]. These studies have validated that the vasculogenesis and angiogenesis of ECs could be engineered in synthetic hydrogels with a suitable stiffness. However, there is still a considerable gap for the optimal efficiency of EC outgrowth (i.e., the growth duration and robustness of vasculature) between such engineered elastic hydrogel and native ECM or its derivatives, which calls for the in-depth understanding of the mechanosensing cues involved in native ECM.

Besides stiffness, accumulating evidence recently shows that natural ECM networks cross-linked by physical interactions have dynamic remodeling behaviors because of the simultaneously rapid association and dissociation of non-covalent cross-links[16–18]. When natural ECM is subjected to traction (i.e., strain) generated by cells, the matrix networks tend to undergo permanent displacement or deformation, the so-called mechanical plasticity[19]. The plastic ECM networks can be easily remodeled and rearranged in response to cells, facilitating complex cellular behaviors, including spreading, proliferation, migration, and differentiation[20,21]. For example, hydrogels with mechanical plasticity exhibit a significant impact on the migration and spreading of tumor cells[22,23]. Although the role of matrix plasticity in

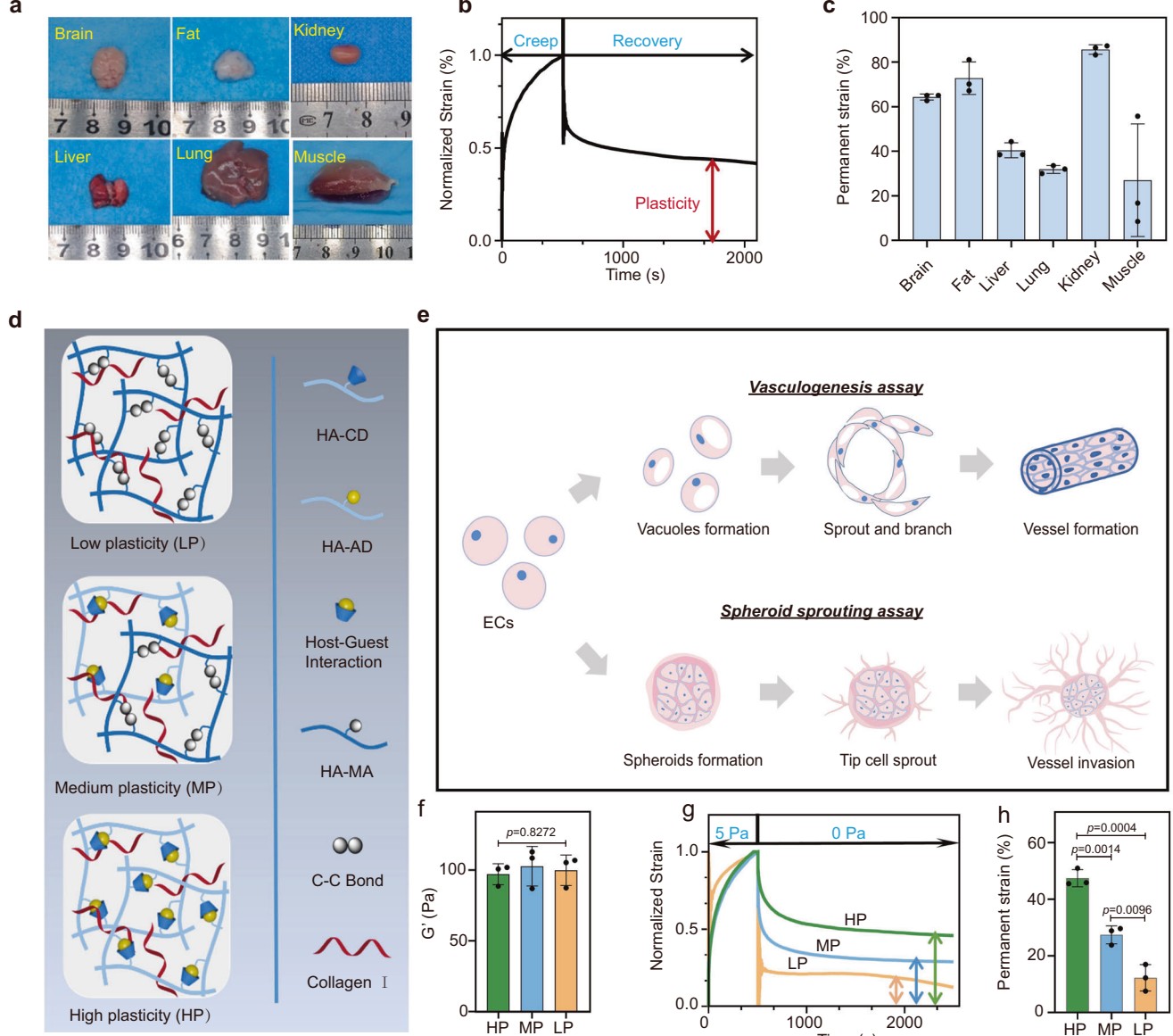

**Fig. 1 | Characterization of mechanical plasticity in vascular-enriched soft tissues and collagen-HA hydrogels with tunable mechanical plasticity independent of stiffness for EC outgrowth in vasculogenesis and spheroid sprouting.** **a** Images of the samples with capillaries-enriched tissues including brain, fat, kidney, liver, lung, and muscle. **b** Representative creep and recovery tests of liver tissues after removing the initial stress. **c** Degree of plasticity in the tested tissues extracted from a mouse of the indicated age (*n* = 3 samples per group). **d** Schematic illustration of the interpenetrating hydrogel networks of high plasticity (HP) based on host-guest physical bonds and low plasticity (LP) based on UV covalent bonds, as well medium plasticity (MP) based on hybrid interactions. **e** Schematic depicts the process of EC outgrowth in vasculogenesis and spheroid sprouting. **f** The *G'* of HP, MP, and LP hydrogels (*n* = 3 samples per group). **g** Representative creep and recovery tests of HP, MP, and LP hydrogels after removing the initial stress of 5 Pa. **h** Permanent strain reflected the degree of plasticity of HP, MP, and LP hydrogels from the creep and recovery tests (*n* = 3 samples per group). All the data are presented as mean values ± SEM. One-way analysis of variance (ANOVA) and two-tailed Student's t-tests are used to assess statistical significance.

tuning individual cell behaviors has been recently well recognized, more attention should be paid to how matrix plasticity modulates EC outgrowth in vasculogenesis and angiogenesis, which not only includes matrix remodeling of individual ECs but also the cell-to-cell crosstalks. However, it remains challenging to engineer hydrogels with tunable mechanical plasticity independent of stiffness and bioactivity for EC outgrowth. In addition, while the mechanosensing signal of integrin, focal adhesion kinase (FAK), and/or cell contractility has been proven to be activated in response to matrix stiffness for tuning ECs assembly[18,24,25], there is still limited understanding of the underlying molecular pathways that transduce the matrix plasticity on ECs.

In this work, we developed hydrogels with tunable mechanical plasticity independent of stiffness, with which we assessed the role of matrix plasticity in mediating EC outgrowth in both vasculogenesis and spheroid sprouting. More specifically, we developed dynamic/covalent cross-linking strategies to engineer hyaluronic acid (HA) hydrogel networks with controllable plasticity, interpenetrating with bioactive collagen I (Col I) fibers. We demonstrated that individual ECs in hydrogels with high plasticity upregulate integrin clusters and stable focal adhesions (FAs), facilitating cell contractility and thus promoting complex vascular assembly. Counterintuitively, we further illustrated that the enhanced cell contractility as induced by the high plasticity of the hydrogels destabilizes intercellular adherens junction, preventing EC invasion. These findings establish that striking a balance of cell contractility in hydrogels with medium plasticity can facilitate both sequential EC migration and vessel stability.

## Results

### Vascular-enriched native tissues exhibit typical mechanical plastic behavior

To define the relevance of tissue plasticity, we first characterized the mechanical plasticity of vascular-enriched soft tissues using a rheometer. Fresh brain, fat, kidney, liver, muscle, and lung tissues are dissected from mice and subjected to rheological testing (Fig. 1a). All these tissues exhibit irrecoverable deformation after removing mechanical compressive loading (Fig. 1b), indicating their typical mechanical plasticity, though the degree of plasticity in terms of permanent strain varies according to different tissue types (Fig. 1c). Among them, vascular flourishing kidneys exhibit substantial plasticity, with a degree of plasticity exceeding 80%. In addition, we found that tissues boasting a high modulus tend to have reduced plasticity, while softer tissues display a higher propensity to permanent deformation after removing the force. However, the relationship between modulus and plasticity is not merely linear, which is also modulated by various factors such as ECM composition, structural design, fiber orientation, and the nature of tissue intercellular interactions. These results establish the relevance of mechanical plasticity to the native microenvironment of ECs.

### Interpenetrating hydrogel networks enable independently tunable mechanical plasticity

To elucidate the effect of mechanical plasticity on EC outgrowth in vitro, it is essential to tune the plasticity of the hydrogels independently of other properties (e.g., stiffness). To achieve this, we designed hydrogels with different plasticity but the same initial stiffness and polymer ingredients using a strategy of dynamic/covalent composite cross-linking. More specifically, we selected HA and Col I as the main backbones. Collagen, as one of the major structural proteins of blood vessels, which has been recognized to provide adhesion sites of RGD (Arg-Gly-Asp) and biodegradable sites of MMP (matrix metalloproteinase)-sensitive peptides for EC outgrowth[26–28]. HA, as a unique glycosaminoglycan widely existing in native ECM[29,30], plays a major role in promoting endothelialization and interacts with several EC surface receptors (e.g., CD44)[31–33]. Host-guest interaction is a bioorthogonal non-covalent reaction that can undergo rapid dynamic exchange

under physiological conditions. Considering their more predictable and reproducible properties based on the precisely defined stoichiometry (1 host + 1 guest complex at each cross-link)[34], several previous studies have developed host-guest HA hydrogel networks with different dynamics but similar stiffness by mixing various pairs of host-guest complexation (with different binding kinetics)[35], however, their adjustable range of dynamic features tuned by this approach is narrow since both of the complexations are consisted of physical host-guest cross-links. To address this, a purely elastic network constructed by fully covalent cross-links was used as static control network in this study. Specifically, we employed the guest-host interactions between 1-adamantaneacetic acid grafted HA (HA-AD) and cyclodextrin grafted HA (HA-CD) as dynamic cross-links for constructing the plastic networks of the hydrogels (Supplementary Fig. 1a). The PBS (pH 7.4) solutions of HA-AD and HA-CD were vortexed at AD:CD = 1:1 and the hydrogel was immediately formed within several seconds. In contrast, the methacrylate modified HA (HA-MA) was covalently cross-linked by photoinitiated polymerization for constructing the purely elastic networks (Supplementary Fig. 1b). The lithium phenyl (2,4,6-trimethylbenzoyl) phosphinate (LAP) as a photoinitiator that can initiate the polymerization under blue light (400–500 nm) with higher initiation efficiency and lower cytotoxicity compared to UV[36,37]. The structures of HA-AD, HA-CD, and HA-MA were determined by NMR (Supplementary Fig. 1c–e). The dynamic HA hydrogels cross-linked by host-guest interactions between HA-AD and HA-CD possessed the typical shear-thinning ability and injectability of dynamic networks (Supplementary Fig. 2a, b).

We developed three groups of hydrogels with varying mechanical plasticity, including high-plastic (HP) hydrogels consisting of purely host-guest networks, medium-plastic (MP) hydrogels consisting of host-guest/UV covalent (1.5:0.5) composite networks and low-plastic (LP) hydrogels consisting of purely UV covalent networks respectively (Fig. 1d). Another important feature in natural ECM of ECs is the fibrillar architecture, which is critical for 3D vasculogenesis and angiogenesis[12,38]. However, the previously reported dynamic tunable HA hydrogel cross-linked by host-guest interactions only forms nanoporous networks and lacks microscale fibrillary. In this study, we introduced the collagen interpenetrated into the above HA networks to mimic the natural ECM of ECs. The concentration of Col was fixed at 0.2 wt% and the concentration of the HA in every Col-HA hydrogel was kept identical at 2.0 wt% throughout the following experiments. In addition, collagen fiber structures in the HP, MP, and LP hydrogels exhibit no significant differences (Supplementary Fig. 2c–e). The HP, MP, and LP hydrogels will be used as scaffolds for EC outgrowth to study their effects on both vasculogenesis and angiogenesis.

As shown in Fig. 1e, vasculogenesis is the de novo formation of primitive vascular networks by individual ECs assembly[39]. The process of vasculogenesis, including vacuole formation[7,40], sprouting, and branching[2,41] can be simulated by encapsulating individual ECs in an engineered hydrogel system[42,43]. In comparison, angiogenesis is initiated from the pre-existing vessels[44,45]. The outgrowing vessels are spearheaded by specialized tip ECs, followed by stalk ECs[46,47]. The spheroid sprouting assay has been widely used for mimicking the angiogenesis process in vitro[48]. The tip cells that sprout from the spheroid lead the invading direction of the ECs[49,50].

To eliminate the effects of matrix stiffness difference on following cell experiments, the mechanical storage moduli ($G'$) of all hydrogel groups were adjusted around ~100 Pa with no significant difference (Fig. 1f), which has been demonstrated to be suitable for ECs spread and growth[51,52]. We then quantified the loss tangent ($G''/G'$) from the rheological frequency sweep (Supplementary Fig. 2f), which shows the decreasing trend of viscosity of the HP, MP, and LP hydrogels due to the gradually reducing proportion of dynamic cross-links (Supplementary Fig. 2g). We have also tested the stress relax behaviors of the HP, MP, and LP hydrogels. As expected, the relax curve of HP hydrogel

 

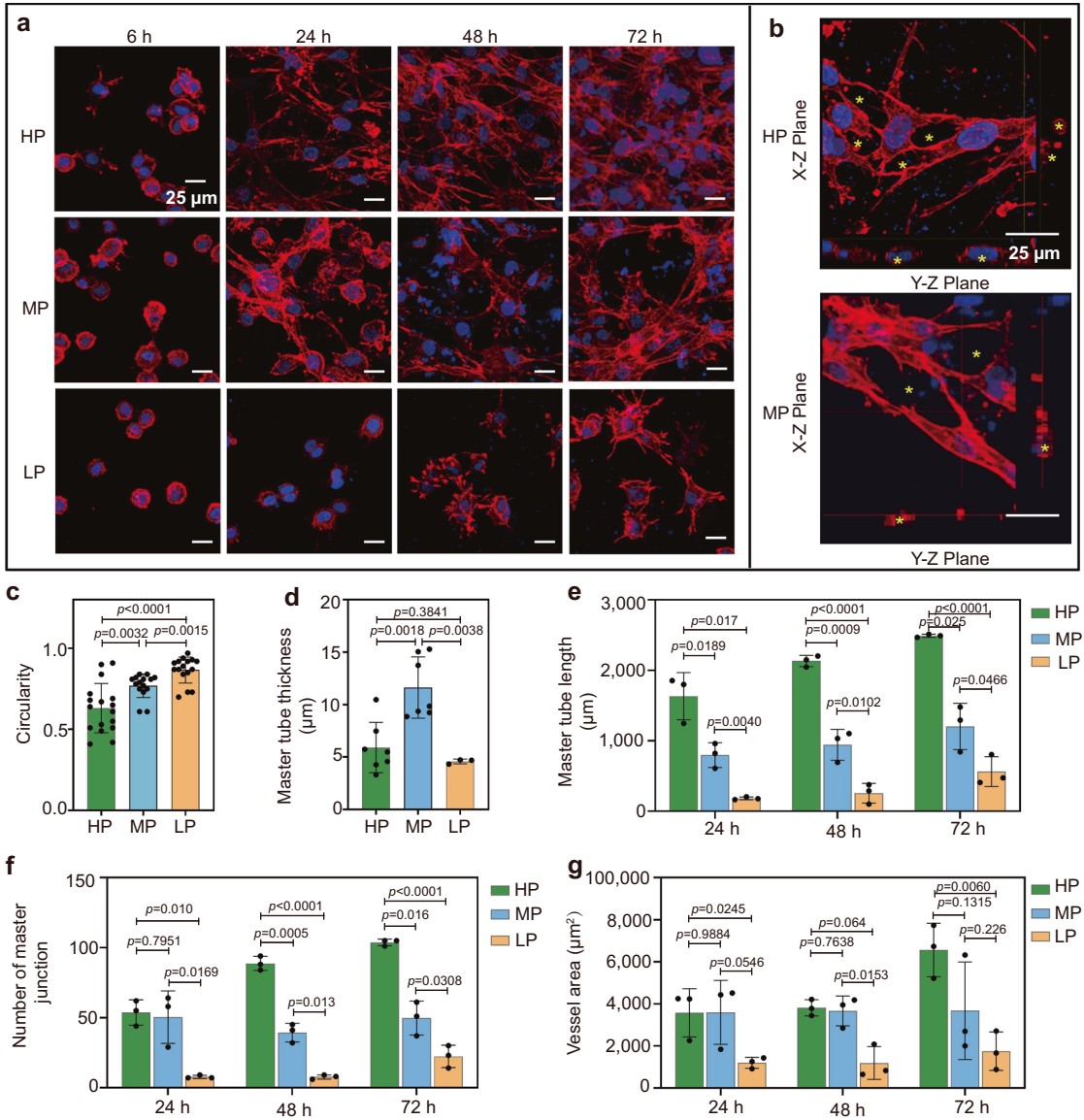

**Fig. 2 | Hydrogels with increasing plasticity promote sprouting and branching of encapsulated endothelial cells (ECs). a** Representative confocal immuno-fluorescent images show morphology changes of encapsulated ECs in HP, MP, and LP hydrogels along 72 h in incubation (F-actin in red and nuclei in blue). Scale bar: 25 μm. **b** Representative orthogonal images of luminal structures (indicated with asterisks) in HP and MP hydrogels after 72 h in incubation (F-actin in red and nuclei in blue). Scale bar: 25 μm. **c** The circularity of ECs in HP, MP, and LP hydrogels after 6 h in incubation (from left to right $n = 17, 15, 16$ cells). **d–g** Quantitative analysis of vascular tube formation of 24 h, 48 h, and 72 h in HP, MP, and LP hydrogels including **d** mean tube thickness (from left to right $n = 7, 7, 3$), **e** master tube length ($n = 3$), **f** the number of master junctions ($n = 3$), and **g** vessel areas ($n = 3$). All the data are presented as mean values ± SEM from three independent experiments. Two-tailed Student's t-tests are used to assess statistical significance.

begins to drop down within 10 s while the curve of LP hydrogel maintains flatly over 600 s (Supplementary Fig. 2h). The quantitative analysis of the half times of stress relaxation also demonstrates that the hydrogels with more dynamic cross-links exhibit faster stress relax rates (Supplementary Fig. 2i). To further evaluate the degree of mechanical plasticity of these hydrogels, a rheological creep recovery test was applied by loading constant initial stress of 5 Pa for 500 s. The stress was subsequently removed, and the resultant creep recovery curves were obtained after normalization (Fig. 1g). The degree of plasticity of the hydrogels was quantified as a percentage of the ratio of the remaining irrecoverable strain to the maximum initial strain. The HP hydrogel exhibits about 50% plasticity, while the MP and LP hydrogels show 25% and 10% plasticity, respectively (Fig. 1h). As a result, the prepared HP, MP, and LP hydrogels offer significantly different plasticity degrees but equally elastic modulus as well as

biopolymer ingredients, providing platforms for the subsequent cellular studies of ECs vasculogenesis and angiogenesis.

## Hydrogel plasticity promotes 3D vascular morphogenesis through mature FA formation and cell contractility

To investigate the role of matrix plasticity on the kinetics of ECs morphogenesis, we encapsulated and tracked human umbilical vein endothelial cells (HUVECs) in the HP, MP, and LP hydrogels at the timepoints of 6 h, 24 h, 48 h, and 72 h, respectively (Fig. 2a, b). The following sprouting, branching and complex vasculature formed with featured lumens were sequentially tracked at each of timepoints, and the 72 h is selected as the maximum timepoints since the comprehensive vascular networks have been observed in HP hydrogels, which collapse or shrink with further prolonged culture over this timepoint (Supplementary Fig. 3). We observed that the ECs in the HP hydrogels

spread more than that in the MP hydrogels, while most ECs in the LP hydrogels remain rounded at 6 h (Fig. 2c). The following sprouting, branching and complex vasculature formed with featured lumens were observed in both the HP and MP hydrogels at 72 h (Fig. 2b). We also compared the characteristics of the vessels formed along with the culture times and found that the lumens are largest in the MP hydrogel (Fig. 2d), while other vessel indicators (e.g., tube length, number of junctions, and vascular covered area) are significantly enhanced with increasing hydrogel plasticity (Fig. 2e–g). In addition, to assess the cell proliferation variances across the hydrogel groups, we used the EdU assay. The results, presented in Supplementary Fig. 4, distinctly highlight the diminished proliferation capacity of cells in LP hydrogels relative to their counterparts in HP and MP hydrogels. This phenomenon can be explained by the plastic remodeling capabilities of HP and MP hydrogels, which create ample spatial allowances conducive for cell mitosis, thereby facilitating greater proliferation. These results suggest that vascular morphogenesis in 3D matrix can be regulated by hydrogel plasticity.

We next examined the molecular pathways by which ECs sense the mechanical plasticity of hydrogels to regulate vascular network formation. ECs interact with mechanical cues from the matrix through FAs and the plasma membrane. To verify the pathways, we investigated the interactions between the integrins of ECs and the hydrogel networks. The clustering of integrin β1 significantly augments with increased hydrogel plasticity, as shown by immunofluorescence analysis (Fig. 3a, b). We then quantified the protein levels of integrin β1 expressed by ECs encapsulated in the HP, MP, and LP hydrogels through Western blot (WB) analysis (Fig. 3c). The quantitative analysis of the WB results demonstrated that the expression of integrin β1 is significantly higher in the HP hydrogels compared with that in the MP and LP hydrogels (Fig. 3d). These results indicate that high hydrogel plasticity promotes the integrin β1expression and thus enhances cell-matrix interactions.

FAK is a key component of the integrin-triggered signal transduction pathway[53, 54], while FAK activation and tyrosine phosphorylation (p-FAK) depend on integrins bound to extracellular signals in a variety of cell types[54]. Since there are different levels of integrin aggregation and expression in the HP, MP, and LP hydrogels, we sought to determine if FAK is activated by hydrogel plasticity. We found that the expression of p-FAK increases significantly with increasing hydrogel plasticity, as shown by immunofluorescence analysis (Fig. 3e, f). Moreover, WB analysis confirmed that both FAK expression and phosphorylation levels of FAK are higher in the HP and MP hydrogels compared with that in the LP hydrogels (Fig. 3g–i). These findings indicate that the mechanical plasticity of hydrogels promotes the formation and phosphorylation of FAK in the encapsulated HUVECs during vasculogenesis.

In addition, the expression of FAK/p-FAK, which acts as a signal transducer between the matrix and cellular cytoskeleton, is closely related to cell contractility. As a result, activation of p-FAK was shown to enhance cell contractility mediated by the myosin light chain phosphorylation (p-MLC)[55,56]. p-MLC operates differently based on its subcellular localization, dictating varied aspects of cell functions and behaviors. The p-MLC primarily influences gene transcription and regulation within the nuclear confines, while it becomes instrumental in modulating cell contraction and cytoskeletal remodeling in the cytoplasm[57,58]. Therefore, we analyzed the distribution and expression of p-MLC in HP, MP, and LP hydrogels. As shown in Fig. 3j–l, the expression of p-MLC is mainly localized in the nuclei of cells in the LP hydrogels with the lowest overall p-MLC expression, which indicates the lowest p-MLC expression in the cytoplasm in LP hydrogels. On the contrary, the highest expression level of p-MLC and its minimal nuclear distribution in HP hydrogel suggest that p-MLC is predominantly expressed in cytoplasm of the cells, resulting in the strongest cell contractility and thus promoting cell spreading. The corresponding

protein levels of p-MLC in the HP, MP, and LP hydrogels were further confirmed by WB analysis (Fig. 3m), which are upregulated with increasing hydrogel plasticity (Fig. 3n). These results indicate that high hydrogel plasticity enables individual ECs to aggregate more integrin clusters, as well as adhesion patches that are stably attached to the matrix to release cellular contractility and promote complex vascular assembly.

Stable intercellular junctions are mediated by vascular endothelial cadherin (VE-CAD), which is highly expressed in the adhesion junctions of the vascular endothelium and is connected to the cytoskeleton by intracellularly mediated proteins[59]. VE-CAD also plays an important role in regulating cell-cell interactions and vascular lumen formation[60,61]. β-catenin (β-cat) binds to the cytoplasmic domain of VE-CAD to form a complex and interacts with the actin cytoskeleton to stabilize intercellular adhesion[62]. Therefore, we first examined the expression levels of β-cat of ECs encapsulated in HP, MP, and LP hydrogels. Both immunofluorescence and WB analysis confirmed that ECs in the MP hydrogels exhibit significantly higher expression of β-cat compared to that in the HP and LP hydrogels (Fig. 4a–d). In addition, although the expression of VE-CAD examined by immunofluorescence shows no significant difference among the hydrogels with different plasticity (Supplementary Fig. 5), we found from the WB analysis that the level of VE-CAD is upregulated in the MP hydrogels (Fig. 4e, f), which was further confirmed by quantitative RT-PCR analysis (Fig. 4g). These results indicate that the β-cat expression aligns with the VE-CAD expression trend and demonstrate that within HP hydrogels, heightened EC contractility triggers the activation of focal adhesion kinase, resulting in dissociation of β-cat from VE-CAD mediated adherens junctions, subsequently disrupting intercellular junctions.

To further confirm the proposed mechanism, we applied a motor-clutch model to simulate EC contractility-mediated mechanical perception of the response to plasticity. Our model contains two viscous elements, including $\eta_1$ for simulating the viscosity of reversible deformation and $\eta_2$ for simulating the degree of irreversible deformation of the substrate(Fig. 4h). The relationship between EC contractility and the plasticity of the matrix was probed via the motor-clutch models that we developed previously[63]. Specifically, we consider the stretching clutch as an integrin that is activated when the clutch elongation exceeds a certain level, thus realizing the motor-clutch model. The activated integrin then promotes the phosphorylation of FAK molecules (Fig. 4i). Subsequently, FAK activation induces the p-MLC and VE-CAD signaling axis, thus enabling mechanotransduction[24, 64,65]. As the degree of plasticity is positively correlated with $\eta_1/\eta_2$, with the increase of $\eta_2$, the irreversible deformation decreases, i.e., the hydrogel plasticity decreases (Fig. 4j). To study the impact of substrate remodeling on cell adhesion, we represented local ligand density changes using different bond association rates. Specifically, our model posits a linear increase in clutch on-rate, characterized as the product of ligand density and true binding rate, with substrate displacement. We simulated the degree of plasticity of the matrix by varying the value of $\eta_2$ (0.1–100). The simulations show that cell adhesion length augments with $\eta_2$ (the irreversible deformation) (Fig. 4k). The adhesion length is positively correlated with cell contractility in the model[63,65]. In addition, we found that when $\eta_2$ is too large ($\eta_2 = 100$), intercellular connections between cells are inhibited because the matrix cannot remodel, while when $\eta_2$ is too small ($\eta_2 = 0.1$), cells cannot form perinuclear forces due to the mobility of the matrix, which also disrupts some of the intercellular connections (Fig. 4l). All these results are consistent with our experimental observations (Figs. 3 and 4e–g). To further verify that VE-CAD is partially disrupted in the HP matrix ($\eta_2 = 0.1$), we inhibited partial cell contractility of HP as 0.1 (-), which leads to an increased VE-CAD expression, reaching the level of MP hydrogel ($\eta_2 = 1$) (Fig. 4m).

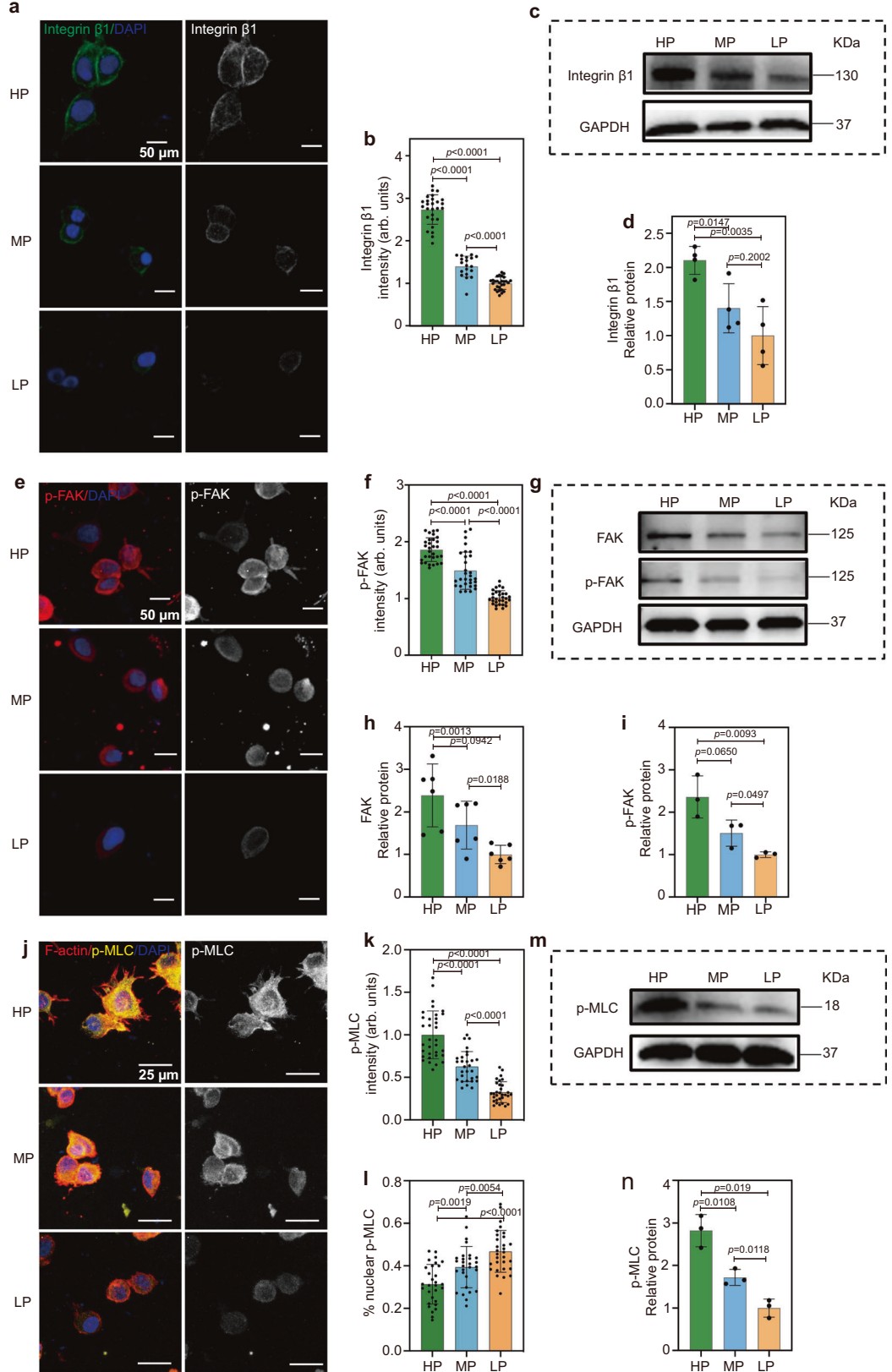

**Hydrogels with medium plasticity facilitate the longest ECs invasion distance and maximum branches in spheroid sprouting**

To investigate how EC angiogenesis is affected by the mechanical plasticity of hydrogels, we fabricated multicellular spheroids of HUVECs, which were encapsulated into the HP, MP, and LP hydrogels and tracked for motile behaviors of ECs with protruded filopodia for 24 h (Fig. 5a, b). Since the role of tip cells is to initiate angiogenesis, the sprouting from EC spheroids is used as an indicator of tip cell formation[66]. We then examined the sprouting of tip cells from each spheroid in the HP, MP, and LP hydrogels and observed that EC spheroid sprouting is enhanced with increasing hydrogel plasticity (Fig. 5c). As the tip cells located at the leading edge of the vascular

**Fig. 3 | Hydrogels with high plasticity promote focal adhesion (FA) formation in encapsulated ECs. a, b** Representative immunofluorescent images and quantification of the normalized intensity of integrin β1 (integrin β1 in green, nuclei in blue) (from left to right $n = 27, 19, 28$ cells from biological triplicate). Scale bar: 50 μm. **c, d** Protein expression levels of integrin β1 determined by Western blotting of ECs encapsulated in different plastic hydrogels and quantified show increased expression of the integrin of ECs encapsulated in hydrogels with increased network plasticity ($n = 4$ tests). **e, f** Representative immunofluorescence images and quantification ($n = 30$ cells for each group from biological triplicate) of the normalized intensity of p-FAK show increased p-FAK activation of ECs encapsulated in hydrogels with increasing network plasticity (p-FAK in green, nuclei in blue). Scale bar: 50 μm. **g** Protein expression levels of FAK (**h**) ($n = 6$ tests) and p-FAK (**i**) ($n = 3$ tests)

determined by Western blotting of ECs encapsulated in different plastic hydrogels show increased expression of the integrin of ECs encapsulated in hydrogels with increased network plasticity. **j, k** Representative immunofluorescent images and quantification of the normalized intensity of p-MLC show enhanced cell contractility of ECs encapsulated in hydrogels with increasing network plasticity (F-actin in red, p-MLC in green, nuclei in blue) (from left to right $n = 33, 29, 31$ cells for each group from biological triplicate). Scale bar: 25 μm. **l** Corresponding quantification of the percentage of nuclear localization to overall protein levels (from left to right $n = 30, 29, 30$ cells for each group from biological triplicate). **m, n** Protein expression levels of p-MLC determined by Western blotting of ECs encapsulated in different plastic hydrogels ($n = 3$ tests). All the data are presented as mean values ± SEM. Student's t-tests are used to assess statistical significance.

sprouts extend long and form dynamic filopodia to direct the migration, we next quantified the length of the filopodia in the hydrogels with different plasticity. The filopodia length per tip cell in the HP and MP hydrogels is 2-fold longer than that in the LP hydrogels (Fig. 5d). However, it is interesting to note that compared to the HP hydrogels, the tip cells in the MP hydrogels exhibit a longer invasion distance and more branches (Fig. 5e, f).

Emphasizing the significance of matrix plasticity, primarily its permanently remodeling ability in EC outgrowth, we incorporated FITC labeled Col I in the HP hydrogel formulation and observed the enrichment of Col I fibers around the cells. Specifically, the ECs were found to permanently rearrange and remodel the networks because of the matrix plasticity (Supplementary Fig. 6a), and tension-induced Col I fiber alignment ultimately leads to Col I enrichment along the branching axis, as can be observed by the EC spheroid sprouting assay (Supplementary Fig. 6b). To further mimic the remodeling of the surrounding matrix by contraction of cells, cyclic loading tests were also performed. The cyclic stress-strain curves have exhibited the plastic-mechanical response of the dynamic HP hydrogel networks (Supplementary Fig. 6c). Specifically, when the hydrogel networks are subjected to cyclic loading forces that mimic cell contraction, they exhibit a nonlinear response. Furthermore, due to the plasticity of the hydrogel, a strain memory effect occurs during cycling[67] (Supplementary Fig. 6d).

Moreover, integrin β1 and FAs of ECs are downregulated with decreasing hydrogel plasticity as shown by immunofluorescence (Supplementary Fig. 7a–d). One possible explanation is that the enhanced contractility of ECs as induced by the stable FAs in the HP hydrogel destabilizes intercellular adhesion junctions, thus preventing the expression of VE-CAD and weakening subsequent stalk cell migration followed by the tip cells. To test this hypothesis, we impeded the FAs and contractility of the ECs in the HP hydrogels by inhibiting FAK using a small molecule inhibitor (at 100 nM) (Fig. 5g). As a result, sprouts from EC spheroids in the HP hydrogels treated with FAK inhibitor invade longer distances with more branches than the untreated controls (Fig. 5h, i). In contrast, in the MP hydrogels, treatment with blebbistatin (BLE), a myosin inhibitor, weakens the sprouting and invasive capabilities of EC spheroids (Supplementary Fig. 8a–c). Furthermore, by employing BV9, a VE-CAD blocking antibody[68], to inhibit VE-CAD in ECs within MP hydrogels (as shown in Fig. 5j), we observed a significant decline in the sprouting and invasion abilities of ECs from the spheroids within BV9 treated environments compared to untreated MP hydrogels (Fig. 5k, l). These results highlight the pivotal role of VE-CAD in endorsing EC spheroid sprouting and also confirm the findings from our previously mentioned mathematical model. The above findings suggest that a balance of cellular adhesion and contractility can be achieved in moderately plastic hydrogels, thereby promoting orderly ECs migration and vascular stability.

### Hydrogels with medium plasticity promote angiogenic effects in vivo

To assess the effects of hydrogel plasticity on angiogenic capacity in vivo, we performed subcutaneous hydrogel implantation in mice

(Supplementary Fig. 9a). For this, hydrogel discs of the HP, MP, and LP samples mixed with SDF-1α and VEGF were implanted under the skin of mice, following an established procedure[69,70]. After 7 days, the samples were explanted and proceeded to sectioning and staining. We found that microvessels invade into all hydrogels (Supplementary Fig. 9b, Fig. 6a, b). From the immunofluorescence image analysis, the percentage of perfusable vessels tended to decrease with decreasing hydrogel plasticity (Fig. 6c). We further quantified the total number of vessels and vascular area from the immunohistochemical images, which are significantly higher in the MP hydrogels compared with that in the HP and LP hydrogels (Fig. 6d, e), consistent with the ECs invasion of spheroid sprouting.

In addition, mean vessel size measured as the distribution of vessel diameters is an important index to quantify angiogenesis[71,72]. To characterize the size and distribution of the vessels, we measured the diameters of all vessels in the hydrogel explants and observed that there are also larger vessels in the MP hydrogels than in the HP and LP hydrogels (Fig. 6f), suggesting that MP hydrogels support effective angiogenesis. Moreover, there are a greater proportion of mature and functionalized vessels in the MP hydrogels (with lumen diameter >6 μm: mammalian red blood cell diameter) than in the HP and LP hydrogels (Fig. 6g). However, the expression of integrin β1 is upregulated with increasing hydrogel plasticity as shown by immunofluorescence (Supplementary Fig. 10a, b), which is consistent with the in vitro experiments (Fig. 3a), demonstrating the same mechanism of overenhanced cell-matrix interactions in vivo. In addition, we performed smooth muscle alpha-actin (α-SMA) staining that can determine the extent of vessel maturation across the different plastic samples (Supplementary Fig. 11a). The number of α-SMA positive vessels is significantly increased in the MP hydrogels, but appears sparse in the HP and LP hydrogels, demonstrating more mature and stabilized vessels in the MP hydrogels (Supplementary Fig. 11b). The in vivo and in vitro experiments corroborated each other and further confirmed the boosting effects of MP matrix on angiogenic ability.

## Discussion

Cells in a non-remodeling three-dimensional(3D) microenvironment often lose their function due to limited space, thus dynamic matrix remodeling is vital for promoting 3D cell behaviors. Although the stress relaxation and plasticity of matrix cannot be decoupled since both of them are derived from the hydrogel network dynamics, the matrix plasticity represents the permanent state after rearrangement of the cross-links by cyclic cell contractions, directly reflecting the ability of dynamic remodeling, which has been reported as a central driver for cell spreading and migration[22,23,67]. However, the assembly of vasculature demands complex physiological processes to occur not only between cells and matrix but cells and neighboring cells. EC outgrowth begins with integrin binding with matrix and proceeds with cell-to-cell interplay both in vasculogenesis and angiogenesis[73–75]. To understand the role of matrix plasticity in EC outgrowth, a tailored hydrogel system is required that can decouple mechanical plasticity from other physical properties (e.g., stiffness), and also allows the

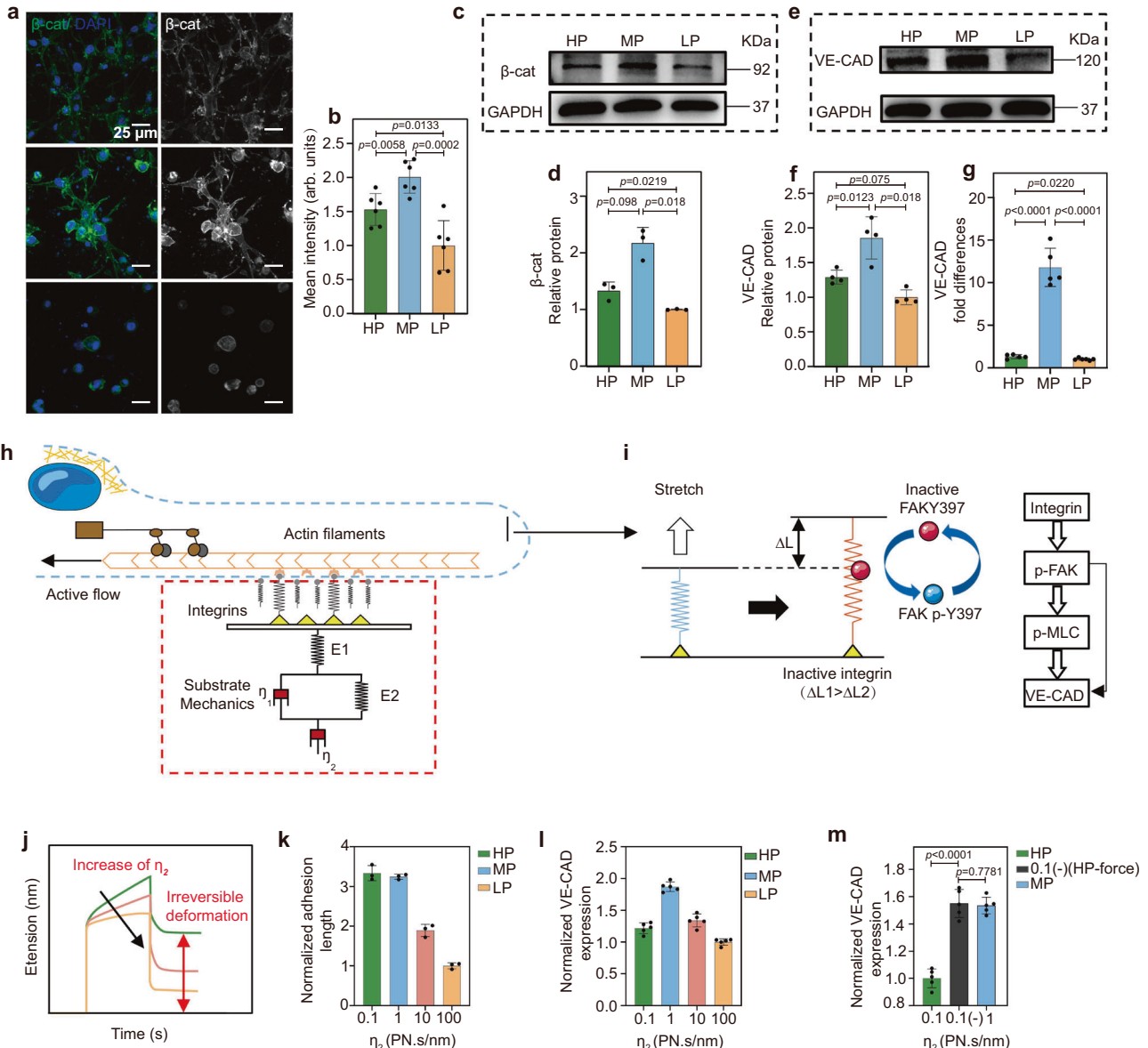

**Fig. 4 | Experimental and mathematical simulations show that hydrogel plasticity mediates cell contractility and adherens junctions. a** Representative immunofluorescent images of β-cat of HUVECs encapsulated in HP, MP, and LP hydrogels. (β-cat in green, nuclei in blue). Scale bar: 25 μm. **b** Quantification of the normalized intensity of β-cat (*n* = 5 tests). **c, d** Protein expression levels of β-cat determined by Western blotting of ECs encapsulated in different plastic hydrogels (*n* = 3 tests). **e, f** Protein expression levels of VE-CAD were determined by Western blotting of ECs encapsulated in different plastic hydrogels (*n* = 4 tests). **g** mRNA expression levels of VE-CAD determined by RT-PCR of ECs encapsulated in different plastic hydrogels (*n* = 6 samples). **h, i** In the motor-clutch model, active integrins (i.e., bonded clutches with stretched lengths over a certain threshold) can enhance the level of FAKY397 phosphorylation at cell adhesions. This mechanochemical reaction is related to the number of active integrins and inactive FAKY397 molecules. **j** The mechanical parameter of $\eta_2$ represents the level of irreversible deformation of the matrix. **k** Simulation results of cell adhesion length in the different plastic matrices (*n* = 3 simulations). **l** Simulated results for VE-CAD levels in the different plastic matrices (*n* = 5 simulations). **m** Simulation results of VE-CAD expression under conditions of inhibition of cell contractility in the HP matrix (*n* = 5 simulations). All the data are presented as mean values ± SEM. Student's t-tests are used to assess statistical significance.

physiological process of vasculature formation. However, existing hydrogel systems do not support both a multicellular tissue formation and an independently tunable mechanical plasticity. Although plenty of dynamic HA hydrogels cross-linked by host-guest interactions have been developed for tracking cell fates[76], their narrow tunable ranges and lack of fibrillar structures limit their applications in studying EC behaviors of vasculogenesis and angiogenesis. In our work, we established a hydrogel system capable of tuning mechanical plasticity, independent of stiffness, by using Col-HA hydrogel systems (HP, MP, and LP hydrogels), with which we unraveled the mechanical plasticity-

mediated responses of ECs during vascular vasculogenesis and angiogenesis.

Taken together, our data reveal that the mechanical plasticity of hydrogels facilitates matrix remodeling by ECs while excessive plasticity has detrimental effects on adherens junction stability between ECs, consistent with our mathematical simulations. Specifically, in vasculogenesis, ECs undergo vacuolization followed by synchronized sprouting and branching to form tubular structures. Due to the high plastic remodeling ability, we found that the HP hydrogels promote stable integrin-RGD cluster of ECs, recruitment of p-FAK with an

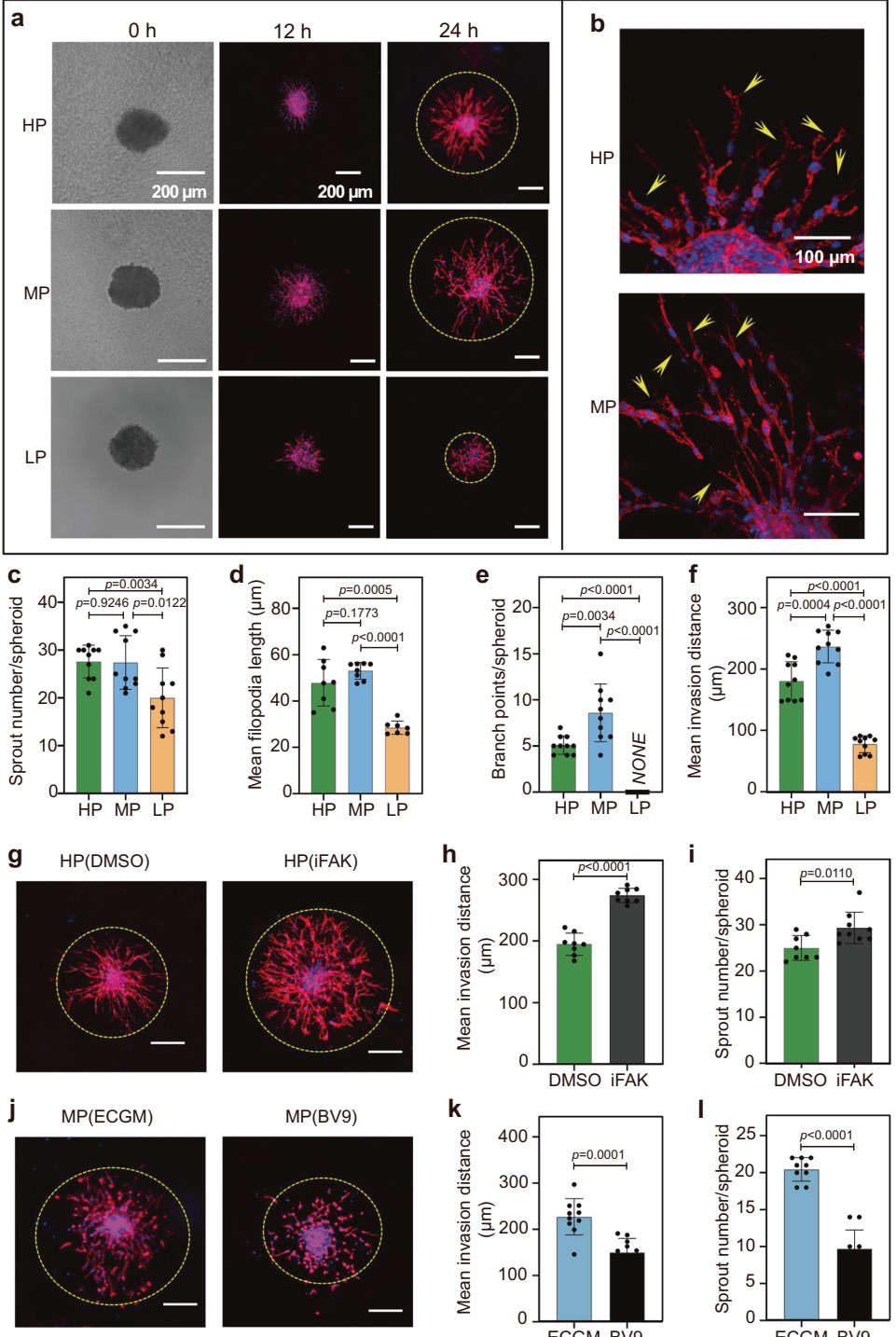

**Fig. 5 | Hydrogels with medium plasticity induce the EC spheroid sprouting.**
**a** Representative confocal immunofluorescent images show the invasive capacity of encapsulated EC spheroids in HP, MP, and LP hydrogels across 24 h in incubation (F-actin in red and nuclei in blue). Scale bar: 200 μm. **b** Representative immuno-fluorescent image of filopodia (indicated with arrows) in HP and MP hydrogels after 24 h in incubation (F-actin in red and nuclei in blue). Scale bar: 100 μm.
**c**–**f** Quantitative analysis of invasive capacity after 24 h of encapsulated EC spheroids in HP, MP, and LP hydrogels including **c** sprout number per spheroid (*n* = 10), **d** filopodia length (from left to right *n* = 8, 8, 7 spheroids), **e** branch points per spheroid (*n* = 10), and **f** mean invasion distance (*n* = 10 spheroids). **g** Representative immunofluorescent images show that low concentration FAK inhibitor promotes ECs invasion in HP hydrogels. **h**, **i** Quantitative analysis of the invasive capacity of EC spheroids encapsulated in HP hydrogels by using FAK inhibitors including **h** mean invasion distance (*n* = 8 spheroids), **i** sprout number per spheroid (from left to right *n* = 8, 9 spheroids). **j** Representative immunofluorescent images show that VE-CAD inhibitor (BV9) inhibited ECs invasion in MP hydrogels. Scale bar: 200 μm.
**k**, **l** Quantitative analysis of the invasive capacity of EC spheroids encapsulated in MP hydrogels by using BV9 including **k** mean invasion distance (*n* = 10 spheroids), **l** sprout number per spheroid (from left to right *n* = 9, 10 spheroids). All the data are presented as mean values ± SEM from three independent experiments. Two-tailed Student's t-tests are used to assess statistical significance.

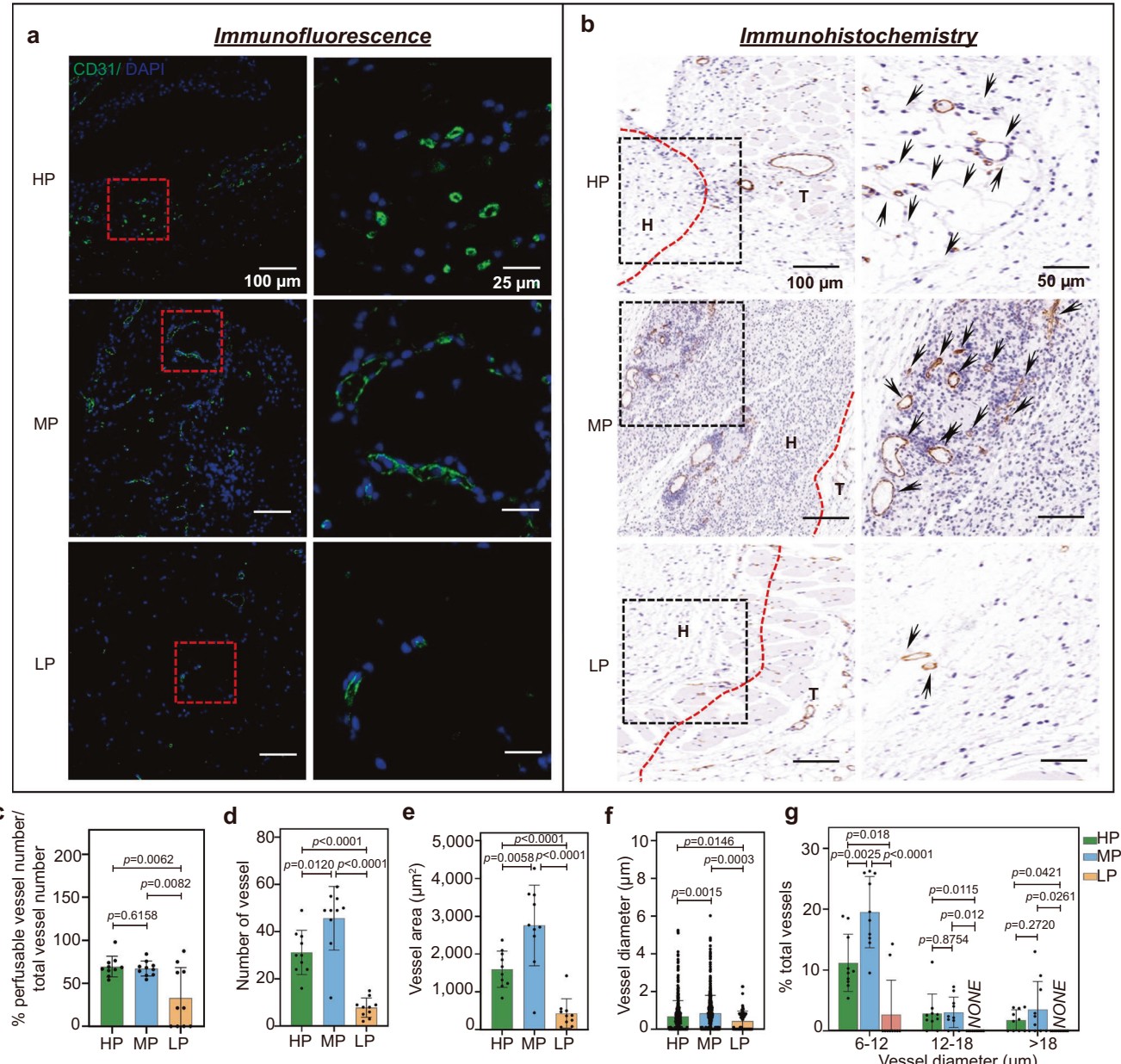

**Fig. 6 | Hydrogels with medium plasticity promote in vivo angiogenic effects.** **a**, **b** The HP, MP, and LP hydrogels were subcutaneously implanted into the backs of mice (*n* = 5 per hydrogel), and the implants were retrieved after 7 days, including **a** representative immunofluorescent images of microvessels infiltrating into hydrogel stained with CD31 (CD31 in green, and nuclei in blue, Scale bars: 100 and 25 μm) and **b** immunohistochemistry images of microvessels stained positive for human CD31 (black arrowheads, Scale bars: 100 and 50 μm). **c**–**g** Quantitative analysis of the ability of HP, MP, and LP implants to induce vascular network

formation including **c** the percent of perfusable vessels (*n* = 10), **d** the number of vessels for CD31 positive (*n* = 10), **e** the vessel covered areas (*n* = 10), **f** the average blood vessel diameter (from left to right *n* = 480, 597, 77 data points from 5 mice. Each mice selected two images, and the diameters of all blood vessels in the two chosen images were recorded and analyzed) and **g** the size distribution of the luminal cross-sectional area (*n* = 10). All the data are presented as mean values ± SEM. In (**c**–**g**), *n* = 5 mice with a total of 10 images were analyzed per group. Student's t-tests are used to assess statistical significance.

increase in p-MLC expression and cell contractility, leading to an expansive vasculature formation (Fig. 7). However, the down-regulation of VE-CAD and the adherens junction destabilization results in smaller lumens than those in the MP hydrogels, due to the over-enhanced contractility as induced by HP hydrogel networks which lead to the dissociation of β-cat from VE-CAD. VE-CAD has been reported to play an essential role in regulating the function of vascular permeability and its reduction results in partial failure of normal vascular lumen formation and leakage of vessels[60,61]. Recent studies also demonstrated that cell contractility should be well balanced, the shift of which to either side can disrupt the integrity of intercellular adhesion junctions by dissociating the linker protein and weakening the

expression of VE-CAD[65,77,78]. In correlation with EC spheroid sprouting, cell contractility is lower in the MP hydrogels, leading to an increase in VE-CAD, which results in the stable adhesion junction between the stalk and tip ECs, allowing the longest vessel invasion distance (Fig. 7). Moreover, inhibitions of adhesions and contractility of ECs in HP hydrogels promote ECs invasion, further confirming such balance contraction-mediated mechanisms. Simultaneously, a larger proportion of mature and functional vessels were observed in the MP hydrogels for in vivo angiogenic model, in accordance with the in vitro assay. The LP hydrogels are demonstrated to result in abrogation of all cell signalings, which ultimately inhibits both vasculogenesis and angiogenesis of ECs. These studies demonstrate that matrix plasticity

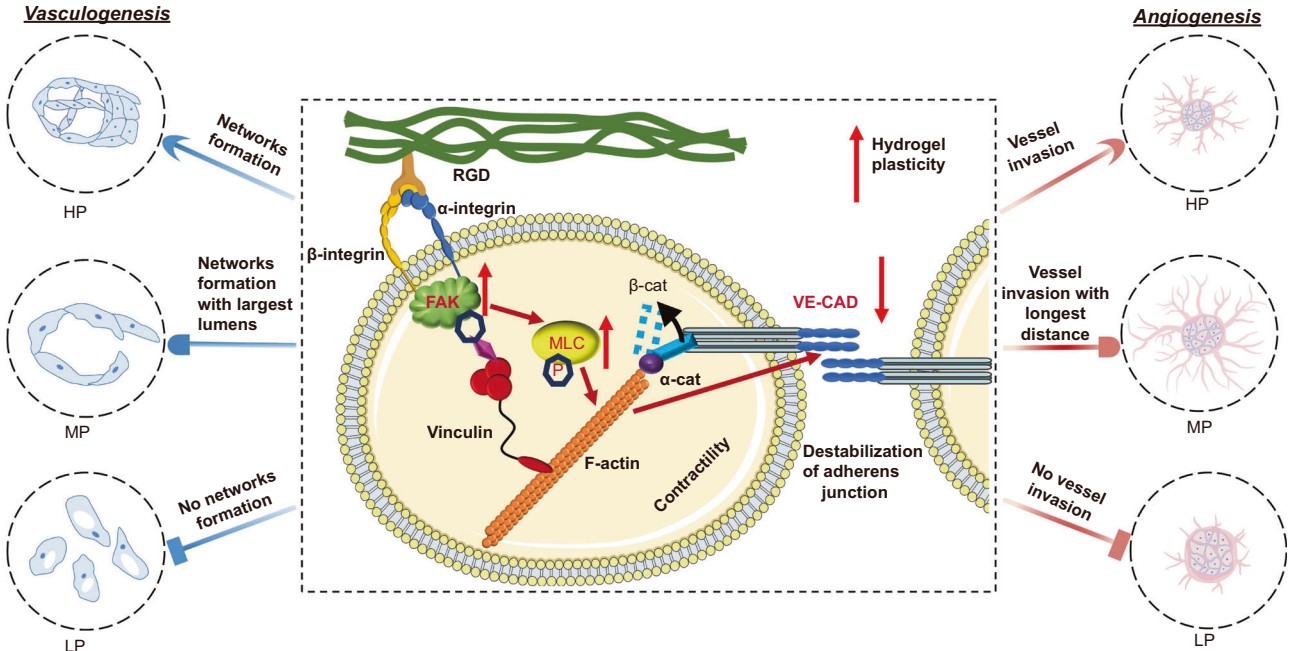

**Fig. 7 | Proposed molecular pathway of EC outgrowth in response to plastic hydrogels.** Hydrogels with high plastic networks enable the rapid formation of integrin clusters and p-FAK activation in a stiffness-independent manner. Activated FAK further contributes to an increase in p-MLC expression and cell contractility, which dissociates β-cat from the VE-CAD and downregulates its expression, disrupting the integrity of intercellular adhesion junctions. By this molecular pathway, ECs are found with the largest lumens formed in MP hydrogels for vasculogenesis and present the longest invasion distance in MP hydrogels by spheroid sprouting manner, demonstrating the matrix plasticity-mediated ECs behaviors.

is a key mediator for regulating EC outgrowth. Our findings evolve our understanding of the underlying mechanosensing mechanisms of cell-matrix and cell-cell interactions and also help optimize the biomaterial design for tissue engineering and blood vessel regeneration, with implications for basic and translational approaches.

## Methods
### Animals
The study involving animals was reviewed and approved by the Ethics Committee of Xi'an Jiaotong University. Animal experiments were carried out in compliance with internationally recognized guidelines, and there were no instances of misuse or ethical violations.

### Chemicals
Sodium hyaluronic acid (HA) was purchased from Shanghai Yuanye Bio-Technology. Dowex 50W×8 resin was purchased from Energy Chemical. Ethyl ether and acetone were purchased from Sinopharm Group. Anhydrous dimethylsulfoxide (DMSO) (anhydrous, ≥99.9%) was purchased from J&K Scientific. Methacrylic anhydride (MA, 94%), Sodium hydroxide (NaOH, 96%), Tetrabutylammonium hydroxide solution (40% (wt/vol) in water), 1-Adamantaneacetic acid (98%), 4-(Dimethylamino)pyridine (DMAP, 97%), Di-tert-butyl dicarbamate chloride (BOC$_2$O, 98%), p-Toluenesulfonyl chloride (PTSC, 99%), Acetonitrile (>99%), (Benzotriazol-1-yloxy)tris(dimethylamino)phosphonium hexafluorophosphate (BOP, 98%), β-Cyclodextrin (98%), Ammonium chloride (99%), N,N-Dimethylformamide (DMF, 99%), 1,6-hexanediamine (HDA, 99.5%) were purchased from Aladdin (China). DMSO-d6 (99.9 atom % D) and D$_2$O (99.9 atom % D) were purchased from Sigma-Aldrich.

### Synthesis of HA-AD, HA-CD, and HA-MA
HA-AD was prepared by esterification of 1-adamantane acetic acid with HA-TBA mediated by BOC2O/DMAP (Supplementary Fig. 1a). To synthesize HA-TBA, HA (1 g) was dissolved in 50 ml deionized water followed by 3 g Dowex 50 W×8 and continue stirring for 30 min at RT. The HA liquid was then collected by vacuum filtration. The obtained solution was adjusted to pH=7 using tetrabutylammonium hydroxide solution and lyophilized. To synthesize HA-AD, 1 g of HA-TBA, 0.82 g of 1-adamantane acetic acid, and 0.13 g of DMAP were added to a flask and purged with nitrogen, followed by the addition of 50 ml of anhydrous dimethylsulfoxide using a syringe. When HA-TBA was fully dissolved, 0.12 ml of BOP was added and the reaction was allowed to stir for 24 h at 45 °C. After the reaction, the solution was purified by dialysis against DI water for 10 days. The obtained solution was adjusted to pH 7 using NaOH and lyophilized. Proton nuclear magnetic resonance ($^1$H NMR) was employed to verify the successful grafting. The appearance of new proton peaks (1.5–1.9 ppm) in the $^1$H NMR spectrum indicates that AD is successfully coupled to the HA backbone. The grafting rate of HA-TBA with AD (38.1%) was determined by the integration of the ethyl polymorph (12H) relative to the HA (Supplementary Fig. 1d). HA-CD was prepared by the amidation reaction of β-CD-HDA with HA-TBA mediated by BOP (Supplementary Fig. 1a). To synthesize β-CD-HDA, we first need to prepare 6-o-monotosyl-6-deoxy-b-cyclodextrin (CD-Tos). To synthesize CD-Tos, 20 g of cyclodextrin was dissolved in 125 ml of deionized water. The solution was subsequently cooled in an ice bath. Then, 4.2 g of PTSC (dissolved in 10 ml of acetonitrile) was added, and the reaction was vigorously stirred for 2 h at room temperature. After the reaction, the crude product was repeatedly suspended and centrifuged using deionized water, acetone, and ether. Subsequently, the precipitate was evaporated overnight under vacuum pressure to remove organic residues, resulting in a dried CD-Tos product. To synthesize CD-HDA, 5 g CD-Tos was added to a three-neck round-bottom flask and purged with nitrogen, followed by the addition of 25 ml of DMF using a syringe. When the mixture was fully dissolved, 17 ml of HDA was added and the reaction was allowed to stir for 20 h at 80 °C. After the reaction, the reaction solution was precipitated by cold acetone, followed by repeated washing of the crude product with acetone and ether. Subsequently, the precipitate was evaporated for

2 days under vacuum pressure to get the dried CD-HDA product. To synthesize CD-HA, 2.5 g of HA-TBA, and 3 g CD-HDA were added to a flask and purged with nitrogen, followed by the addition of 125 ml of anhydrous dimethylsulfoxide using a syringe. When the mixture was fully dissolved, 1.06 g of BOP (dissolved in 20 ml of anhydrous DMSO was added and the reaction was allowed to stir for 3 h at room temperature. After the reaction, the solution was purified by dialysis against DI water for 6 days. The obtained solution was adjusted to pH = 7 using NaOH and lyophilized. The appearance of new proton peaks (1.2–1.9 ppm) in the $^1$H NMR spectrum indicates that CD is successfully coupled to the HA chains (Supplementary Fig. 1e). To determine the grafting rate of HA-TBA with CD (26.1%), the hexane linker (12H) was integrated relative to the N-acetyl singlet of HA (3H). The methacrylated HA (HA-MA) was prepared by the reaction of HA with methacrylic anhydride (Supplementary Fig. 1b, c). To synthesize HAMA, 1 g HA was dissolved in 50 ml of deionized water and the solution was cooled in an ice bath. The MA solution was added dropwise with stirring at 4 °C. NaOH was added dropwise for 9 h to maintain the pH between 8–9. After reaction, the solution was purified by dialysis against DI water for 6 days. The obtained solution was adjusted to pH 7 using NaOH and lyophilized.

### Preparation of the HP, MP, and LP hydrogels

The specific method of mixing the elastic and plastic systems and interspersing Col I in the above network is as follows: to prepare HP hydrogels, the stock solutions of HA-CD, HA-AD, and collagen were uniformly mixed in a stoichiometric ratio of 1:1 for AD and CD. The self-assembly of the host-guest group can be completed in seconds, and the hydrogel needs to be placed at 37 °C and incubated for 30 min to trigger the polymerization of Col I. MP hydrogels were prepared by uniformly mixing stock solutions of HA-CD, HA-AD, Col I, HA-MA with high modification and photoinitiator LAP solutions. Specifically, HA-CD stock solution (3.6 wt% in PBS, pH 7.4) was first mixed with Col I system (including 1X PBS, 0.2 mol NaOH, ECGM, 1 wt% Col I stock solution), HA-MA stock solution (2 wt% in PBS, pH 7.4) and LAP stock solution (5 wt% in PBS, pH 7.4). Then HA-AD stock (2.4 wt% in PBS, pH 7.4) was mixed with the above mixture by a luer lock. The hydrogel was first allowed to polymerize at 37 °C for 30 min and then irradiated with blue light (405 nm) for 8 s to trigger covalent cross-linking. LP hydrogels were prepared by uniformly mixing HA-MA solution with low modification, LAP stock solution, and collagen solution. The mixture was placed in the 37 °C incubator for 30 min and then irradiated under blue light (405 nm) for 6 s. The Col I was interpenetrated to the above HA networks at a fixed concentration of 0.2 wt% and the concentration of the HA in every sample was kept identical and constant at 2.0 wt% throughout the following experiments. In addition, the photoinitiator content in both LP and MP hydrogels was 0.05 wt%.

### Mechanical tests

Rheological tests were performed using a rotational rheometer (Anton Paar MCR 302) at 37 °C. Fresh heart, lung, liver, kidney, muscle, and brain tissues were dissected from mice. Tissues were cut into 8 mm diameter discs and placed in a cold PBS buffer, then the sample temperature was recovered to room temperature prior to testing. Creep and recovery tests were performed for characterizing the plasticity of the tissue. A constant shear force (15 Pa) was applied for 500 s and then withdrawn the stress and held for 2000 s. Continuous recording of time and strain changes throughout the test.

The hydrogels were prepared as 15 mm diameter and 1.5 mm thickness discs to characterize their rheological properties. Storage modulus ($G'$) and loss modulus ($G''$) were measured in a frequency scan in the range of 0.1–10 Hz (1% strain). Similar to the measurement of tissue plasticity, a constant shear force (5 Pa) was applied to samples

for 500 s and then withdrawn the stress and held for 2000 s. In addition, stress relaxation measurements of hydrogels were performed by time sweep tests at a constant initial strain of 1%. The relaxation rate was quantified as the time for the initial stress to half of its original value. Cyclic loading with gradually increasing amplitude was performed to analyze the plastic behavior of HP hydrogels (Supplementary Fig. 5c).

### Cell culture and 3D encapsulation

HUVECs were purchased from Cyagen Biosciences (HUVEC-90011) and cultured in Endothelial cell growth medium (ECGM, Promocell) and 1% Pen/Strep (Life Technologies, Solarbio). The purchased second-generation cells were inoculated in culture flasks and cultured to the sixth generation for subsequent experiments. For 3D culture, cell suspensions were mixed into the hydrogel precursor solution before hydrogel formation, which were encapsulated at a final density of 2,000,000 cells/mL. The hydrogels were placed in a humidified chamber at 37 °C and 5% $CO_2$ for 30 min, and then the growth medium was added to each gel and replaced every 24 h. The morphology of HUVECs in hydrogels was followed by optical microscopy and confocal microscopy.

### Cell proliferation (EdU assay)

The EdU cell proliferation assay was conducted as per the prescribed manufacturer's protocol, utilizing cells at a density of 2,000,000 cells/mL. Cells were treated with a 10 μM concentration of the EdU reagent (Beyotime, C0071S) and incubated for 2 h, allowing for effective labeling. Post-incubation, the cells underwent three thorough PBS washes, followed by fixation using a 4% paraformaldehyde solution for 30 min. Thereafter, they were permeabilized with 0.5% Triton X-100 for 20 min. In the subsequent phase, cells were exposed to the click-reaction reagent for 60 min, shielded from light and at ambient temperature. Concluding the procedure, DAPI was used to stain the nuclei.

### Generation of endothelial cell spheroids

HUVEC spheroids with a certain number of cells ($n = 500$) were generated as previously reported[44]. In brief, HUVECs were uniformly mixed in ECGM containing 0.25% w/v carboxymethyl cellulose and pipetted on a Petri dish for hanging drop formation. Sterile PBS was added to the dishes to maintain a moist environment, and then placed in the incubation chamber overnight to form spheroids.

### The spheroid sprouting assay

For the sprouting assay, spheroids were embedded into HP, MP, and LP precursor solutions. After hydrogel formation, culture medium was added and the spheroids were allowed to sprout for 12 h and 24 h. In vitro sprouting was quantified using Image J software which measured the number of sprouts, the invasion distance of sprouts, the length of filopodia, and the branch points per spheroid. 8–12 spheroids from each group were analyzed. For inhibition treatments, corresponding inhibitors were added after the spheroids were encapsulated for 12 h, using the following reagents: blebbistatin (40 μM, Selleck, S7099), FAK inhibitor (100 nM, PF-562271 (Med ChemExpress, USA)), BV9 (50 μM, Santa Cruz Biotechnology, sc-52751).

### Collagen labeling

Fitc-collagen was prepared in the same way as previously reported in the literature[79]. In brief, the FITC solution was slowly added to the Col I solution and allowed to react at 4 °C for 8 h. Subsequently, $NH_4Cl$ was added, and the reaction was terminated at 4 °C for 2 h. The final reaction solution was dialyzed at 4 °C for 3 days (molecular weight cutoff: 6000–8000 Da), followed by freeze-drying. Finally, the freeze-dried product was dissolved in 0.1% glacially acetic acid for subsequent use.

## Immunofluorescence (IF)

Hydrogels containing vascular networks or EC spheroids were fixed using 4% paraformaldehyde at 37 °C for 30 min and washed with PBS. For staining, the samples were permeabilized with 0.5% Triton-X (Sigma, USA) for 20 min and washed with PBS, then 5% bovine serum albumin was added for 2 h. The samples were treated with primary antibody in antibody diluent solution overnight at 4 °C, then washed with PBS. Samples were then incubated for 2 h with the corresponding secondary antibody and Phalloidin. Finally, staining with DAPI for 10 min before using confocal microscopy. Primary antibodies/reagents used are as follows: anti-integrin β1 (1:1000 for 3D vascular network and 1:200 for EC spheroids, Santa Cruz Biotechnology, sc-9970), anti-phospho-FAK (1:1000; Thermo Fisher Scientific, 700255), anti-pMLC (1:1000; Cell Signaling Technology, 3675S), anti-β-cat (1:500, Santa Cruz Biotechnology, sc-7963), anti-VE-CAD (1:500 for 3D vascular network and 1:200 for EC spheroids, Abcam, ab33168), anti-paxillin (1:500 for 3D vascular network and 1:200 for EC spheroids, Abcam, Y113), and anti-vinculin (1:500 for 3D vascular network and 1:200 for EC spheroids, Abcam, EPR8185). Alexa Fluor 488 goat anti-rabbit secondary antibody (1:1000, Abcam, ab150077) and Alexa Fluor 488 goat anti-mouse secondary antibody (1:1000, Abcam, ab150113) were used for counterstain. Alexa 565-labeled Phalloidin (1:1000, Sigma, 94072) and DAPI (1:5000, Sigma, D9542) were used to visualize the F-actin and label nucleus, respectively.

## RT-PCR analysis

Total RNA was harvested from HUVECs encapsulated in HP, MP, and LP hydrogels using TRIzol. The extracted RNA was reverse transcribed to cDNA using the RevertAid First Strand cDNA Synthesis Kit (Thermo Fisher Scientific, k1622). Real-time PCR was performed using SYBR Premix Ex Taq II (Accurate biology, AG11701) on the Real-time PCR instrument (ABI, America). The primer sequences used were as follows: GAPDH: fwd 5′-GAGGGTCTCTCTCTTCCTCTTGT-3′, rev 5′-CTCCTC TGACTTCAACAGCGACA-3′; VE-CAD: fwd 5′-GGCTCAGACATCCACAT AACC-3′, rev 5′-CTTACCAGGGCGTTCAGGGAC-3′.

## Western blot

The milled hydrogels were lysed on ice using RIPA buffer (Beyotime, P0013J) containing 1% protease inhibitor for 30 min. After the hydrogels were lysed, SDS buffer was added and boiled at 100 °C for 5 min, and then loaded onto 10% SDS gel. Total protein extracts were separated by electrophoresis and the proteins were then transferred to polyvinylidene difluoride membranes (Millipore, USA). The membranes were blocked with fast blocking solution (Beyotime, P0252) for 30 min, followed by the addition of the corresponding primary antibody, and incubated overnight at 4 °C. Then, the membranes were washed with TBST and incubated with secondary antibodies for 2 h at room temperature. The protein signal was probed with a chemiluminescence system (Chemi-Scope 3300 mini, China). Primary antibodies/reagents including anti-GAPDH (1:1000, SAB, 21612), anti-integrin β1 (1:1000, Santa Cruz Biotechnology, sc-9970), anti-FAK (1:1000, Cell Signaling Technology, 71433S), anti-phosphorylation -FAK (1:1000, Cell Signaling Technology, 8556S), anti-pMLC (1:1000, Cell Signaling Technology, 3675S), anti-β-cat (1:1000, Santa Cruz Biotechnology, sc-7963), anti-VE-CAD (1:1000, Abcam, ab33168), goat anti-mouse IgG secondary antibody HRP conjugated (1:5000, SAB, L3032) and goat anti-rabbit IgG secondary antibody HRP conjugated (1:5000, SAB, L3042).

## Motor-clutch model

The cell adhesion on plastic substrate can be modeled by the motor-clutch dynamics as our previous model. In our motor-clutch model, the substrate and integrin molecules are modeled by elastic springs. These molecules spring are stretched by actomyosin with an effective speed of $V_r$,

$$V_r(F_s) = V_0 \left(1 - \frac{F_s}{F_{stall}}\right), \tag{1}$$

$$F_s = K_{link} \sum_{i=1}^{N_{link}} (x_i - x_s), \tag{2}$$

where $V_0 = 120$ nm/s is the base flow rate of actin filaments; $F_{stall} = 100$ pN is the stall force of myosin motors; $N_{link} = 50$ is the number of engaged clutch bonds; $x_i$ (nm) is the elongation that spring $i$; $x_s$ (nm) is the displacement of substrate; $K_{link} = 0.8$ pN/nm is the spring constant. The rupture rate of engaged bond is,

$$k_{off} = k_0 e^{F_i/F_b}, \tag{3}$$

$$F_i = K_{link}(x_i - x_s), \tag{4}$$

where $F_b = 2$ pN is the characteristic rupture force; $k_0$ is the base off-rate. Besides, the effective binding rate increases with bond tension as,

$$k_{on} = k_{on}^0 C_{int}, \tag{5}$$

where $k_{on}^0 = 0.3$ s$^{-1}$ is the base on-rate and $C_{int}$ is the density of integrins with a base value of 300/μm². Based on previous research findings[24], the degree of FAK phosphorylation has a counteractive effect on VE-CAD dissociation. Therefore, this model incorporates an additional component to make hypotheses regarding integrin-adhesion cross-talk, based on existing experimental data. In our model, the VE-CAD activation and inactivation are governed by the following equations:

$$r_{1f} = k_{CAD}[C_{CAD-ina}] \tag{6}$$

$$r_{1r} = k_{CAD-ina}[C_{VE-CAD-a}] \tag{7}$$

where $r_{1f}$ is association rate of VE-CAD for single VE-CAD and $r_{1r}$ is the disassociation rate of VE-CAD for clustered VE-CAD; $[C_{CAD-ina}]$ and $[C_{CAD-a}]$ represent the concentrations of single-state cadherin and clustered cadherin, $k_{CAD-a}$ (0.3 s$^{-1}$) is the association rate of VE-CAD. Based on existing research experiments, we used a linear relation between VE-CAD inactivation rate ($k_{cad-off}$) and cell traction force ($F_s$),

$$k_{cad-ina} = \gamma \frac{K}{(F_s[C_{int}] + K)} \tag{8}$$

where $\gamma$ is the inhibition coefficient ($\gamma = 1$) and $K$ is the Hill coefficient.

## In vivo subcutaneous implantation of hydrogels

The animal studies were approved by Xi'an Jiaotong University, School of Life Science and Technology (Xi'an, China). Hydrogel discs were prepared by dropping 100 μl of hydrogel solution containing SDF-1α (200 ng/ml) and VEGF (50 ng/ml) into disc-shaped molds. C57 mice (6–8-week-old, males) were divided into three groups of HP, MP, and LP hydrogels, with three individuals in each group. The mice were anesthetized with isoflurane gas and fixed on the operating platform. Both flanks were dissected and disinfected. 5 mm incisions were made on flank skin in lateral recumbent position, the hydrogel discs were then implanted subcutaneously on both sides and sutured. After feeding under the same conditions for 7 days (21 °C and 50% humidity), all mice were euthanized, and the hydrogels were dissected out, fixed with 4% PFA to make paraffin sections subsequently.

## Immunohistochemistry

For immunohistochemical analysis, ethanol was used to gradientlly dehydrate paraffin-embedded tissues. Antigen retrieval is then performed and incubated with the corresponding antibody including anti-CD31 (1:500, Aabcam, ab28364) and anti-α-SMA (1:200, Bioss Antibodies, bsm-33187M). For immunofluorescence of Integrin β1 and CD31, anti-rabbit IgG secondary antibody Alexa Fluor 546 conjugate (Invitrogen, 1:500, A11008) was used. For statistical analysis, 10 fields of view at 20x were selected for each group, and the number of vessels and vessel area were quantified using Image J. The short diameter of each vessel was also measured to characterize vessel maturation.

## Statistics

Statistical analysis was performed using GraphPad 7.0. All data are presented as mean ± SD. Two-tailed t-test to determine statistical significance. One-way analysis of variance (ANOVA) was used to carry out multiple comparisons among the three groups. The threshold for statistically significant differences between groups was $p < 0.05$. In materials experiments, multiple samples were tested and the data are presented as mean values ± SEM. For cell experiments, data from multiple trials are displayed as mean values ± SEM. In animal experiments, $n = 5$ mice with a total of 10 images were analyzed per group, and the data are presented as mean values ± SEM.

## Reporting summary

Further information on research design is available in the Nature Portfolio Reporting Summary linked to this article.

## Data availability

All data in this study are available in the manuscript and the Supplementary Information or from the corresponding author upon request. Source data are provided with this paper.

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

## Acknowledgements

This work was financially supported by the National Natural Science Foundation of China (12225208, 12002263) and the Young Talent Support Plan of Xi'an Jiaotong University. Supported by the Natural Science Basic Research Program of Shaanxi (2022KJXX-52), the Shaanxi Young Talent Recruitment Program.

## Author contributions

Z.W., M.L., Y.W., S.G. and F.X. conceived the ideas and designed the experiments. Z.W., M.L., Y.W., Y.X., X.X., D.L., B.C., S.G. and F.X. conducted the experiments and analyzed the data. All authors interpreted data and contributed to the writing of the manuscript.

## Competing interests

The authors declare no competing interests.
