## [Peer Review File · Nature Communications]

Reviewers' Comments:

Reviewer #1:

Remarks to the Author:

The paper by Xu and Gerecht under consideration at Nature Communications look at the mechanical plasticity of the extracellular matrix during endothelial cell activity. The idea of matrix plasticity influencing cell behavior is not a new concept, having been demonstrated to play important roles in numerous contexts as pointed out by the authors (ref 22/23). The authors used a similar approach to these previous works, where different crosslinking approaches could tune the degree of mechanical plasticity independently of bulk modulus. The use of covalent/non-covalent host-guest chemistry provides a unique system where the plasticity and modulus can be controlled. The authors performed several in vitro and in vivo assays to study endothelial cell processes and discovered a role for matrix plasticity in guiding distinct vasculogenic behavior. Overall this is an interesting paper with some promising results. However, I feel that the mechanistic support for the final conclusion is sparse and needs bolstering. The following points should be taken into account:

1. The major point of the paper is in the idea of matrix "mechanical plasticity" and the role this plays in complex morphogenesis like endothelial cell sprouting. The authors demonstrate a materials platform for decoupling stiffness from plasticity. However, it is not clear whether the system here allows decoupling of stress relaxation, a characteristic that has been shown to be highly important in regulating many diverse cell and tissue level activities. The authors need to demonstrate comparable stress relaxation across their HP, MP and LP materials, or otherwise rationalize why stress relaxation is not the central driver behind the observed cellular processes.
2. The authors show how pMHC is increased in the higher plasticity conditions. Why is the p-MHC showing nuclear localization in the low plasticity material relative to the others? Indeed the images in Figure 4A show very low pMHC fluorescence altogether. The role of nuclear pMHC needs to be discussed and how the differences relate to the observed phenotypes.
3. The observation that VE-CAD is higher in the medium plasticity condition is based on the Western result in Figure 4F. However, these blots look identical by eye and the Figure S4 is stated to show no difference but the expression looks higher in these immunofluorescence results for cells in the high and medium plasticity conditions relative to low plasticity. More evidence for VE-CAD regulation on the medium plasticity matrices is needed.
4. Many of the images merely suggest that the cells within the low plasticity material are unhappy with low/no protrusive features. Was there a difference in proliferation between the conditions?
5. The motor-clutch model is a nice addition to the story. I understand the relationships between viscosity and plasticity, and how these might relate to the phosphorylation events underlying contractility. However, it is not clear how these results support the VE-CAD results. Some additional description of the model and how the expression levels of these effectors is driven by the mechanical parameters in the model is essential.
6. The authors state that the beta1 integrin and focal adhesions are downregulated with decreasing plasticity (Figure S5). However, this is not clear in the images at all. I would say that the beta1 integrin is lowest in the medium plasticity condition compared to the others. This is also slightly apparent in the focal adhesion stains. This statement needs to be supported through quantification.
7. The blebbistatin results are suggestive that contractility is influencing the plasticity-sprouting phenotype. However, there are many other pathways perturbed downstream of myosin inhibition. It is confusing why the authors do not attempt to link the VE-CAD results discussed in earlier sections to the sprouting assay. Is there a relationship—as described in the single cell/spheroid work—between the propensity to sprout and the model of matrix plasticity directing contractility and the VE-CAD phenotype? Would blocking VE-CAD in the sprouting assay attenuate the enhancement observed in the medium plasticity materials? This experiment of staining for VE-CAD and blocking this process is critical to supporting the model results.

As it stands, there are numerous pieces of interesting evidence, but most of this is correlative and requires some additional mechanistic work to parse out the interplay between endothelial cells and matrices of variable mechanical plasticity. The following minor points should be taken into consideration.

In Figure 3, B and C should be labelled consistently; they currently flipped in the panel.

Page 3 line 75 remodeling should read remodelling

Page 3 line 76 challenging should read challenging

I encourage a careful read of the manuscript as there are many spelling and grammatical errors throughout.

Reviewer #2:

Remarks to the Author:

This manuscript describes an interesting and well-orchestrated study of how engineered ECM plasticity affects endothelial cell outgrowth and blood vessel formation. The manuscript reads well overall and, with *ex vivo*, *in vitro*, and *in vivo* evaluations, has the potential for impact in understanding how matrix plasticity influences vascularization, with implications for biomaterials designs and fundamental mechanisms. However, there are several points that need clarification for fully appreciating the innovation and potential impact of the presented work as noted below.

1. What plasticity means in the context of the presented work should be briefly addressed in the abstract, if space is available, for the reader more fully appreciate the points being made there.
2. The examination of the plasticity of tissues at the start of the Results (Figure S1) is interesting. It would be worth considering including these data in the main figure if possible. Additionally, how these data compare to prior work in the literature examining the moduli of tissues (e.g., the work of Shelly Peyton, <https://doi.org/10.1371/journal.pone.0204765>) should be addressed in the Results/Discussion.
3. Collagen should be defined as Collagen I on its first mention in the Results. Additionally, how similar or different the presented materials are from those previously reported in the literature should be noted in the Results and further discussed in the Discussion if needed with relevant references – references to published protocols for building block synthesis are noted in the Methods, but what aspects of the materials formulations are established vs. new needs to be clarified. This information is needed for understanding the potential innovations and impact of the presented work in the context of the literature (e.g., are the materials new, is the application of them new, or are both the materials and application new).
4. Use of visible light is noted “rather than traditional UV, avoiding potential cytotoxicity of the reaction.” This point needs a reference or further explanation as there is ample work that supports how low doses of long wavelength UV light are not cytotoxic for many cell types including with the photoinitiator LAP (e.g., the work of Kristi Anseth, <https://www.ncbi.nlm.nih.gov/pmc/articles/PMC2896013/>).
5. 72 hours is utilized for evaluating HUVEC response in “Hydrogel plasticity promotes 3D vascular morphogenesis...” Why this timepoint was selected needs to be explained / justified.
6. What number of biological replicates were used (e.g., number of donors or animals) vs. number of sample replicates should be clarified for both *in vitro* and *in vivo* studies with explanation / justification.
7. It is noted that “We then quantified the expression of Beta1 integrin through Western blot (WB) analysis from the hydrogel (Fig. 3C) and found that the HP hydrogel networks recruit more integrins and enhance cell-matrix interactions.” What is being shown and observed in the figure needs to be further described here. How “recruit more integrins” is to be assessed with the WB, which will show expression of integrins at the protein level but not their spatial localization/recruitment, needs to be clarified or perhaps the statement wording adjusted.
8. The manuscript, especially the abstract and part of the introduction, has many typographical errors. A thorough proofread is needed.

We thank the reviewers for their constructive suggestions and comments, which we are happy to address in the current version of the manuscript. To this end we performed a series of new experiments and analyses. With the help of the reviewers, we are convinced to have improved the manuscript significantly.

Response to Reviewer #1

Q1. *The major point of the paper is in the idea of matrix “mechanical plasticity” and the role this plays in complex morphogenesis like endothelial cell sprouting. The authors demonstrate a materials platform for decoupling stiffness from plasticity. However, it is not clear whether the system here allows decoupling of stress relaxation, a characteristic that has been shown to be highly important in regulating many diverse cell and tissue level activities. The authors need to demonstrate comparable stress relaxation across their HP, MP and LP materials, or otherwise rationalize why stress relaxation is not the central driver behind the observed cellular processes.*

A1. We thank the reviewer for the comments. First, we would like to point out that the plasticity of hydrogel cannot be decoupled from the stress relaxation since they are interrelated features arising from the dynamics of hydrogel networks (refer to Nat. Rev. Mol. Cell Biol. 24, 495 (2023); Nat. Commun. 9, 4144 (2018)). Reinforcing this understanding, we also directly tested the stress relaxation behaviors of the HP, MP, and LP hydrogels. The subsequent results, shown in new data of **fig. S2**, indicate that the relaxation curve of HP hydrogel starts to decline within 10 seconds, in stark contrast to the LP hydrogel, which remains consistent for over 600 seconds (**fig. S2H**). A quantitative analysis of the half-times of stress relaxation further underscores that hydrogels with increased dynamic crosslinks manifest faster stress relaxation rates (**fig. S2I**).

Changes were made: Supplementary information, Page 3-4

Fig. S2. The injectability and shear-thinning of the plastic HA hydrogels, as well as the fibrous structures and the rheological tests of the Col-HA hydrogels.

(H) Stress relaxation curves for HP, MP, and LP hydrogels with normalized stress to initial measurements. (I) Quantification of timescales at which the stress relaxes to half its original value from stress relaxation tests in H.

Furthermore, we agree with the reviewer that it is important to distinguish between stress relaxation and plasticity. While stress relaxation is a time-dependent phenomenon experienced when cells contact with the matrix, plasticity is time-independent, ensuring sustained deformations even after cells disengage from the networks (refer to *Nat. Rev. Mol. Cell Biol.* 24, 495 (2023)). As a result, the matrix plasticity represents the final state after rearrangement of the crosslinks, directly reflecting the ability of matrix remodeling, which is vital for large deformation and migration of the cells, such as the process of endothelial cell (EC) morphogenesis and angiogenesis. In comparison, time-dependent stress relaxation only describes the contacting status between matrix and cell on an unpredictable time scale. The irreversible plasticity of matrix constituted by dynamic networks has been reported as central driver for cell spreading and migration, although the stress relaxation is not decoupled in these systems (refer to *Nat. Commun.* 9, 4144 (2018); *Nat. Commun.* 8, 842 (2017); *Adv. Healthc. Mater.* 10, 2001856 (2021); *Nat. Commun.* 12, 2759 (2021); *Adv. Healthc. Mater.* 2301586, (2023)).

Besides, to strengthen our claim that mechanical plasticity, the permanently remodeling ability of the matrix, plays crucial roles in vasculogenesis and angiogenesis, the FITC labelled Collagen I (Col I) has been used in the preparation of hydrogels. The enrichment of the Col I fibers shown in HP hydrogels demonstrates the permanent rearrangement of the networks by endothelial cells (ECs) (**Fig. S6A**), and the EC spheroid sprouting shown that the tension-induced Col I fiber alignment ultimately leads to Col I enrichment along with the EC branching axis (**fig. S6B**), demonstrating the effects of plastic remodeling of the matrix. To further mimic and confirm the plastic remodeling of the surrounding matrix by contraction of cells, a cyclic loading test was also performed as inspired by the previous study (refer to *Nat. Commun.* 12, 2759 (2021)). The plastic-mechanical response of the plastic hydrogel networks can be clearly observed by the cyclic stress-strain curves (**fig. S6C**). Specifically, the hydrogel

networks exhibit a nonlinear response and a strain memory effect occurs during cyclic loading forces (**fig. S6D**). All these data provided direct evidence to demonstrate the crucial role of permanently plastic remodeling of matrix on cell behaviors.

Changes were made: Supplementary information, Page 8

Fig. S6. Illustrations of the plastic hydrogel networks remodeling by cells.

(A) High-resolution images from a confocal microscope display an increase in Col I density surrounding ECs embedded within the HP hydrogel, alongside adjacent space. Scale bar: 10 μm. (B) Depictions of an EC spheroid embedded in HP hydrogel reveal enhanced Col I density in proximity to the cell, together with an accompanying channel. Color keys: Col I in green, DAPI in blue, and phalloidin in red. Scale bar: 25 μm. (C) Cyclic loading test with increasing amplitude. (D) Cyclic stress-strain of HP hydrogel shows plastic remodeling. During cyclic loading, the strain memory effect can be observed.

To clarify these, we have added new paragraphs of results and discussion in the main text of the manuscript as follows,

In Results section:

Changes were made: Manuscript, Page 9

“We have also tested the stress relax behaviors of the HP, MP and LP hydrogels. As expected, the relax curve of HP hydrogel begins to drop down within 10 s while the curve of LP hydrogel maintains flatly over 600 s (**fig. S2H**). The quantitative analysis of the half times of stress relaxation also demonstrates that the hydrogels with more dynamic crosslinks exhibit faster stress relax rates (**fig. S2I**).”

Changes were made: Manuscript, Page 16

“Emphasizing the significance of matrix plasticity, primarily its permanently remodeling ability in EC outgrowth, we incorporated FITC labeled Col I in the HP hydrogel formulation and observed the enrichment of Col I fibers around the cells. Specifically, the ECs were found to permanently rearrange and remodel the networks because of the matrix plasticity (**fig. S6A**), and tension-induced Col I fiber alignment ultimately leads to Col I enrichment along the branching axis, as can be observed by the EC spheroid sprouting assay (**fig. S6B**). To further mimic the remodeling of the surrounding matrix by contraction of cells, cyclic loading tests were also performed. The cyclic stress-strain curves have exhibited the plastic-mechanical response of the dynamic HP hydrogel networks (**fig. S6C**). Specifically, when the hydrogel networks are subjected to cyclic loading forces that mimic cell contraction, it exhibits a nonlinear response. Furthermore, due to the plasticity of the hydrogel, a strain memory effect occurs during cycling⁶⁷ (**fig. S6D**).”

In Discussion section:

Changes were made: Manuscript, Page 19

“Cells in a non-remodeling three-dimensional (3D) microenvironment often lost their function due to limited space, thus dynamic matrix remodeling is vital for promoting 3D cell behaviors. Although the stress relaxation and plasticity of matrix cannot be decoupled since both of them are derived from the hydrogel network dynamics, the matrix plasticity represents the permanent state after rearrangement of the crosslinks by cyclic cell contractions, directly reflecting the ability of dynamic remodeling, which has been reported as a central driver for cell spreading and migration^{22,23,67}.”

In Method section:

Changes were made: Manuscript, Page 23-24

“Mechanical tests

In addition, stress relaxation measurements of hydrogels were performed by time sweep tests at a constant initial strain of 1%. The relaxation rate was quantified as the time for the initial stress to half of its original value. Cyclic loading with gradually increasing amplitude was performed to analyze the plastic behavior of HP hydrogels (fig.S6C).”

Changes were made: Manuscript, Page 25-26

“Collagen labeling

Fitc-collagen was prepared in the same way as previously reported in the literature⁸¹. In brief, the FITC solution was slowly added to the Col I solution and allowed to react at 4°C for 8 hours. Subsequently, NH₄Cl was added, and the reaction was terminated at 4°C for 2 hours. The final reaction solution was dialyzed at 4°C for 3 days (molecular weight cut-off: 6,000-8,000 Da), followed by freeze-drying. Finally, the freeze-dried product was dissolved in 0.1% glacially acetic acid for subsequent use.”

Q2. The authors show how pMHC is increased in the higher plasticity conditions. Why is the p-MHC showing nuclear localization in the low plasticity material relative to the others? Indeed the images in Figure 4A show very low pMHC fluorescence altogether. The role of nuclear pMHC needs to be discussed and how the differences relate to the observed phenotypes.

A2: We appreciate the reviewer’s comment. First and foremost, we would like to clarify that we mentioned p-MLC (myosin light chain phosphorylation) rather than pMHC throughout the manuscript. The multifaceted roles of p-MLC are based on its localization within cells, which have been previously documented. Specifically, when localized in the nucleus, p-MLC primarily governs gene transcription and regulation (refer to *Nat. Commun.* 9, 2124 (2018)). Contrastingly, within the cytoplasm, its predominant function revolves around the assembly and dissociation of actin, which subsequently facilitates cytoskeletal remodeling, cell contraction, and motility (refer to *FASEB J.* 33(8), 9062 (2019)).

Our observations, as reflected in **Fig. 3J-L**, resonate with these established roles. The expression of p-MLC is mainly localized in the nuclei of cells in the LP hydrogels with the lowest overall p-MLC expression. On the contrary, the highest expression level of p-MLC and its minimal nuclear distribution in HP hydrogel suggest that p-MLC is predominantly expressed in cytoplasm of the cells, resulting in the strongest cell contractility and thus promoting cell spreading.

Following the reviewer's suggestions, we have added the following discussion to explain the role of p-MLC and its relationship with observed phenotypes of cells, and we have also revised the description of **Fig. 3L** in the manuscript.

Changes were made: Manuscript, Page 12-13

“p-MLC operates differently based on its subcellular localization, dictating varied aspects of cell functions and behaviors. The p-MLC primarily influences gene transcription and regulation within the nuclear confines, while it becomes instrumental in modulating cell contraction and cytoskeletal remodeling in the cytoplasm^{57,58}. Therefore, we analyzed the distribution and expression of p-MLC in HP, MP and LP hydrogels. As shown in **Fig. 3J-L**, the expression of p-MLC is mainly localized in the nuclei of cells in the LP hydrogels with the lowest overall p-MLC expression, which indicates the lowest p-MLC expression in cytoplasm in LP hydrogels. On the contrary, the highest expression level of p-MLC and its minimal nuclear distribution in HP hydrogel suggest that p-MLC is predominantly expressed in cytoplasm of the cells, resulting in the strongest cell contractility and thus promoting cell spreading.”

Furthermore, as for the reviewer's concerns about the subdued p-MLC fluorescence in **Fig. 3J**, we have re-did the relevant cell culture and staining experiments to improve our imaging quality. The new images in **Fig. 3J** now elucidate clear distinctions among HP, MP, and LP hydrogels and the corresponding quantitative analysis in **Fig. 3K and L** have also been performed which shows the same trend as before, enhancing our claims.

Changes were made: Manuscript, Page 48-49

Figure 3. Hydrogels with high plasticity promote focal adhesion (FA) formation in encapsulated ECs.

(J, K) Representative immunofluorescent images and quantification of the normalized intensity of p-MLC show enhanced cell contractility of ECs encapsulated in hydrogels with increasing network plasticity (n=30 cells from biological triplicate) (F-actin in red, p-MLC in green, nuclei in blue). Scale bar: 25 μ m. (L) Corresponding quantification of the percentage of nuclear localization to overall protein levels (n = 30 cells from biological triplicates).

Q3. The observation that VE-CAD is higher in the medium plasticity condition is based on the Western result in Figure 4F. However, these blots look identical by eye and the Figure S4 is stated to show no difference but the expression looks higher in these immunofluorescence results for cells in the high and medium plasticity conditions relative to low plasticity. More evidence for VE-CAD regulation on the medium plasticity matrices is needed.

A3: To address the reviewer's concern of the Western result in Fig. 4, we have re-did the Western Blot experiments of VE-CAD by increasing the sample amount for each group. The differences among the hydrogel groups became more pronounced in Fig. 4E and the update quantitative analysis from the blots has also been confirmed in Fig. 4F.

Changes were made: Manuscript, Page 50

Figure 4. Experimental and mathematical simulation show that hydrogel plasticity mediates cell contractility and adherens junctions.

(E, F) Protein expression levels of VE-CAD determined by Western blotting of ECs encapsulated in different plastic hydrogels.

As for the reviewer's concern about the immunofluorescence of VE-CAD, we have carried out the immunofluorescence staining for VE-CAD once again. Our enhanced results and additional statistical analyses verify that the fluorescence intensities among the three groups do not exhibit significant variations. Consequently, we have updated the manuscript with the new representative images for immunofluorescence and appended further statistical data in **fig. S5A, B**.

Changes were made: Supplementary information, Page 7

Fig. S5. Expression of intercellular VE-CAD in HP, MP and LP hydrogels.

(A) Representative immunofluorescent images of VE-CAD of HUVECs encapsulated in HP, MP and LP hydrogels. (VE-CAD in green, nuclei in blue). Scale bar: 25 μm. (B) Quantification of the normalized intensity of VE-CAD. All the data are presented as mean values \pm SEM, $n \geq 3$ biological replicates per group, 'ns' indicates no statistical difference.

In addition, to further address the reviewer's concerns, we have incorporated supplementary evidence related to VE-CAD regulation. This encompasses results from our model studies (as elaborated in Q5), detection of upstream protein expression levels (detailed to Q7), and our exploratory inhibition experiments (detailed in Q7).

Q4. Many of the images merely suggest that the cells within the low plasticity material are unhappy with low/no protrusive features. Was there a difference in proliferation between the conditions?

A4: We are grateful for the reviewer's insights. To assess the differences in cell proliferation across the three hydrogel groups, we have conducted cell proliferation experiments utilizing the EdU assay. The findings from this experiment have been incorporated as **fig. S4** in the manuscript. The data reveals a notably reduced cell proliferation ability in the LP hydrogels in contrast to both HP and MP hydrogels. This observation can be attributed to the plastic remodeling inherent to HP and MP hydrogels, which furnishes the requisite space for cell mitosis, thus fostering enhanced cell proliferation, as supported by the study in *Nat. Mater.*15, 326–334 (2016). We have elucidated this in the manuscript with added data, discussions, and a detailed description of the experimental methodology.

Changes were made: Supplementary information, Page 6

Fig. S4. Cell proliferation ability in HP, MP and LP hydrogels.

(A) EdU assay of cell in HP, MP and LP hydrogels. EdU positive nuclei (labeled with Alexa Fluor 488 azide; green) and Hoechst-stained nuclei of all the cells (blue) were visualized by fluorescence microscopy. Scale bar: 25 μm . (B) Statistical graph of the EdU assay in (A). All the data are presented as mean values \pm SEM, n=5 biological replicates per group, 'ns' indicates no statistical difference, ***P < 0.001, ****P < 0.0001.

In Results section:

Changes were made: Manuscript, Page 10-11

“In addition, to assess the cell proliferation variances across the hydrogel groups, we used the EdU assay. The results, presented in **fig. S4**, distinctly highlight the diminished proliferation capacity of cells in LP hydrogels relative to their counterparts in HP and MP hydrogels. This phenomenon can be explained by the plastic remodeling capabilities of HP and MP hydrogels, which creates ample spatial allowances conducive for cell mitosis, thereby facilitating greater proliferation.”

In Method section:

Changes were made: Manuscript, Page 24-25

“Cell proliferation (EdU assay)

The EdU cell proliferation assay was conducted as per the prescribed manufacturer's protocol. Cells were treated with a 10 μM concentration of the EdU reagent (Beyotime, C0071S) and incubated for 2 hours, allowing for effective labeling. Post-incubation, the cells underwent three thorough PBS washes, followed by fixation using a 4% paraformaldehyde solution for a duration of 30 minutes. Thereafter, they were permeabilized with 0.5% Triton X-100 for 20 minutes. In the subsequent phase, cells were exposed to the click-reaction reagent for 60 minutes, shielded from light and at ambient temperature. Concluding the procedure, DAPI was used to stain the nuclei.”

Q5. The motor-clutch model is a nice addition to the story. I understand the relationships between viscosity and plasticity, and how these might relate to the phosphorylation events underlying contractility. However, it is not clear how these results support the VE-CAD results. Some additional description of the model and how

the expression levels of these effectors is driven by the mechanical parameters in the model is essential.

A5: We sincerely appreciate the insightful suggestions from the reviewer. Our manuscript elucidates the mechanism by which cells sense and react to a plastic substrate, drawing upon the molecular clutch model. The remodeling of such substrates, particularly the resulting integrin clustering due to heightened local ligand density, directly influences cell adhesion as highlighted in *Nat. Mater.* 15, 326–334 (2016). To understand the impact of substrate remodeling on cell adhesion, we represented local ligand density changes by using different bond association rates. Specifically, our model posits a linear increase in clutch on-rate, characterized as the product of ligand density and true binding rate, with substrate displacement. This can be observed in **Fig. 4K**, illustrating that cell adhesion length augments with η_2 (the irreversible deformation).

Shifting focus to **Fig. 4L**, our consideration of VE-CAD levels led us to introduce another model assumption, one centered on the interplay between integrin-adhesion, rooted in current experimental evidence. This underscores the intricate relationship between cell-matrix (integrin-FAK axis) and cell-cell signaling (involving cadherin proteins), both of which collectively regulate a spectrum of cellular activities including adhesion, locomotion, and signal transduction. Case in point: phosphorylated FAK can instigate the dissociation of β -cat and VE-CAD complexes, a result of matrix stiffening-mediated cell contractility as stated in *Adv. Sci.* 22, 2201483 (2022). Furthermore, subsequent studies on FAK’s downstream proteins unearthed the overexpression of proteins like RhoA and ROCK. This surge amplifies cell contractility, triggering VE-CAD dissociation and, consequently, boosting cell permeability, as discussed in *Sci. Rep.* 7,45835 (2017). With these findings in the backdrop, it is deduced that FAK phosphorylation intensity conversely influences VE-CAD dissociation. In our modeling framework, VE-CAD activation and inactivation hinge upon the following equations, detailing the association and disassociation rates:

Description	Reaction	Rate
-------------	----------	------

Association rate of VE-CAD for single VE-CAD	$N_{CAD}^{ina} \rightarrow N_{CAD}^a$	$r_{1f} = k_{CAD}[C_{CAD-ina}]$
Disassociation rate of VE-CAD for clustered VE-CAD	$N_{CAD}^a \rightarrow N_{CAD}^{ina}$	$r_{1r} = k_{CAD-ina}[C_{VE-CAD-a}]$

where $[C_{CAD-ina}]$ and $[C_{CAD-a}]$ represent the concentrations of single-state cadherin and clustered cadherin, k_{CAD-a} (0.3 s^{-1}) is the association rate of VE-CAD. Based on existing research experiments, we used a linear relation between VE-CAD inactivation rate ($k_{cad-off}$) and cell traction force (F_s),

$$k_{cad-ina} = \gamma \frac{K}{(F_s[C_{int}] + K)}$$

where γ is the inhibition coefficient ($\gamma = 1$). Thus, when cell traction increases (high cell traction and low η_2), levels of VE-CAD decreases, as shown in the **Fig. 4L**. When we inhibited the cell traction force, levels of VE-CAD increase again, as shown in **Fig. 4M**. To clarify this, we have added the following description in the manuscript.

In Results section:

Changes were made: Manuscript, Page 14-15

“To study the impact of substrate remodeling on cell adhesion, we represented local ligand density changes using different bond association rates. Specifically, our model posits a linear increase in clutch on-rate, characterized as the product of ligand density and true binding rate, with substrate displacement. We simulated the degree of plasticity of the matrix by varying the value of η_2 (0.1-100). The simulations show that cell adhesion length augments with η_2 (the irreversible deformation) (**Fig. 4K**).”

In Method section:

Changes were made: Manuscript, Page 30-31

“Based on previous research findings⁸², the degree of FAK phosphorylation has a counteractive effect on VE-CAD dissociation. Therefore, this model incorporates an additional component to make hypotheses regarding integrin-adhesion crosstalk, based on existing experimental data. In our model, the VE-CAD activation and inactivation are governed by the follow equations:

$$r_{1f} = k_{CAD}[C_{CAD-ina}] \quad (6)$$

$$r_{1r} = k_{CAD-ina}[C_{VE-CAD-a}] \quad (7)$$

where r_{1f} is the association rate of VE-CAD for single VE-CAD and r_{1r} is the disassociation rate of VE-CAD for clustered VE-CAD; $[C_{CAD-ina}]$ and $[C_{CAD-a}]$ represent the concentrations of single-state cadherin and clustered cadherin; k_{CAD-a} (0.3 s^{-1}) is the association rate of VE-CAD. Based on existing research experiments, we used a linear relation between VE-CAD inactivation rate ($k_{cad-off}$) and cell traction force (F_s),

$$k_{cad-ina} = \gamma \frac{K}{(F_s[C_{int}] + K)} \quad (8)$$

where γ is the inhibition coefficient ($\gamma = 1$) and K is the Hill coefficient.”

Q6. *The authors state that the beta1 integrin and focal adhesions are downregulated with decreasing plasticity (Figure S5). However, this is not clear in the images at all. I would say that the beta1 integrin is lowest in the medium plasticity condition compared to the others. This is also slightly apparent in the focal adhesion stains. This statement needs to be supported through quantification.*

A6: We thank the reviewer for the constructive comments. In response to the concerns raised, we have repeated the immunofluorescence staining, accompanied by a quantitative analysis. To obtain superior-quality immunofluorescence images of the EC spheroids, we have made alterations in the primary antibody incubation. Specifically, the concentration was augmented from 1:500 to 1:200 for anti-paxillin and anti-vinculin, and from 1:1,000 to 1:200 for anti-integrin $\beta 1$. Additionally, the incubation time for the primary antibody was extended from 12 hours to 16 hours. The statistical data have demonstrated the trend that both integrin $\beta 1$ and focal adhesions are generally downregulated with decreasing plasticity of the hydrogels. To clarify this, we have added the data and made modifications to the method section in the manuscript.

Changes were made: Supplementary information, Page 9

Fig. S7. EC spheroids encapsulated in hydrogels with different plasticity.

(A, B) Representative immunofluorescence images and quantification of the normalized intensity of integrin $\beta 1$ of EC spheroids encapsulated in HP, MP and LP hydrogels (F-actin in red, integrin $\beta 1$ in green, nuclei in blue). (C, D) Representative immunofluorescence images and quantification of the normalized intensity of FA of EC spheroids encapsulated in hydrogels of different plasticity (F-actin in red, FA in green, nuclei in blue). Scale bar: 100 μm . All the data are presented as mean values \pm SEM, $n \geq 8$ spheroids from biological triplicate per group, ‘ns’ indicates no statistical difference, $*P < 0.05$, $***P < 0.001$ and $****P < 0.0001$.

Changes were made: Manuscript, Page 26-27

“Primary antibodies/reagents used are as follows: anti-integrin $\beta 1$ (1:1,000 for 3D vascular network and 1:200 for EC spheroids, Santa Cruz Biotechnology, sc-9970), anti-phospho-FAK (1:1,000; Thermo Fisher Scientific, 700255), anti-pMLC (1:1,000; Cell Signaling Technology, 3675S), anti-VE-CAD (1:500 for 3D vascular network and 1:200 for EC spheroids, Abcam, ab33168), anti-paxillin (1:500 for 3D vascular network and 1:200 for EC spheroids, Abcam, Y113), and anti-vinculin (1:500 for 3D vascular network and 1:200 for EC spheroids, Abcam, EPR8185).”

Q7. The Blebbistatin results are suggestive that contractility is influencing the plasticity-sprouting phenotype. However, there are many other pathways perturbed downstream of myosin inhibition. It is confusing why the authors do not attempt to link the VE-CAD results discussed in earlier sections to the sprouting assay. Is there a relationship—as described in the single cell/spheroid work—between the propensity to

sprout and the model of matrix plasticity directing contractility and the VE-CAD phenotype? Would blocking VE-CAD in the sprouting assay attenuate the enhancement observed in the medium plasticity materials? This experiment of staining for VE-CAD and blocking this process is critical to supporting the model results. As it stands, there are numerous pieces of interesting evidence, but most of this is correlative and requires some additional mechanistic work to parse out the interplay between endothelial cells and matrices of variable mechanical plasticity.

A7: We thank the reviewer for pointing these out. Following the reviewer’s suggestion, we have assessed VE-CAD blocking in MP hydrogel by employing the VE-CAD inhibitor BV9. This antibody specifically targets and binds to a distinct region of the VE-CAD molecule, effectively inhibiting its ability to bind to other VE-CAD molecules or adhesion molecules on adjacent cells. Our data from this approach revealed that both the invasion distance and sprouting number in BV9-treated groups are considerably lower in comparison to the untreated MP groups. This underscores a direct association between sprouting and VE-CAD in our sprouting assay and further substantiates the results derived from our model. We have incorporated these data in the manuscript as Fig. 5J-M and also added the related discussion.

Changes were made: Manuscript, Page 17

“Furthermore, by employing BV9, a VE-CAD blocking antibody⁶⁸, to inhibit VE-CAD in ECs within MP hydrogels (as shown in **Fig. 5J**), we observed a significant decline in the sprouting and invasion abilities of ECs from the spheroids within BV9 treated environments compared to untreated MP hydrogels (Fig. **5L, M**). These results highlight the pivotal role of VE-CAD in endorsing EC spheroid sprouting and also confirm the findings from our previously mentioned mathematical model.”

Changes were made: Manuscript, Page 52-53

Figure 5. Hydrogels with medium plasticity induce the EC spheroid sprouting.

(J) Representative immunofluorescent images show that VE-CAD inhibitor (BV9) inhibited ECs invasion in MP hydrogels. Scale bar: 200 μm . (L, M) Quantitative analysis of the invasive capacity of EC spheroids encapsulated in MP hydrogels by using BV9 including (L) mean invasion distance, (M) sprout number per spheroid. All the data are presented as mean values \pm SEM, $n \geq 8$ spheroids from biological triplicate per group, ns indicates no statistical difference, * $P < 0.05$, ** $P < 0.01$, *** $P < 0.001$ and **** $P < 0.0001$.

Moreover, to comprehensively address the reviewer's concerns and authenticate the proposed mechanistic trajectory, we have intensified our focus on the interactions between ECs under varying mechanical plasticity conditions. Recognizing that β -cat is an upstream protein associated with VE-CAD and that it partners with VE-CAD's cytoplasmic domain to stabilize intercellular adhesions (refer to *J Cell Biol.* 129, 203–217, 1995), we have carried out immunofluorescent staining and Western Blot studies for β -cat of ECs encapsulated in HP, MP, and LP hydrogels. Our findings, presented in new data of **Fig. 4A-D**, unexpectedly highlighted that the β -cat expression peaks in MP hydrogels in comparison to its counterparts, aligning with the VE-CAD expression trend. This indicates that within HP hydrogels, heightened EC contractility triggers the activation of focal adhesion kinase, resulting in dissociation of β -cat from VE-CAD mediated adherens junctions, subsequently disrupting intercellular junctions. These insights have been encapsulated and elaborated upon in the manuscript.

Changes were made: Manuscript, Page 13-14

“ β -catenin (β -cat) binds to the cytoplasmic domain of VE-CAD to form a complex and interacts with the actin cytoskeleton to stabilize intercellular adhesion⁶². Therefore, we first examined the expression levels of β -cat of ECs encapsulated in HP, MP and LP hydrogels. Both immunofluorescence and WB analysis confirmed that ECs in the MP hydrogels exhibit significantly higher expression of β -cat compared to that in the HP and LP hydrogels (**Fig. 4A-D**). In addition, although the expression of VE-CAD examined by immunofluorescence shows no significant difference among the hydrogels with different plasticity (**fig. S5**), we found from the WB analysis that the level of VE-CAD is upregulated in the MP hydrogels (**Fig. 4 E, F**), which was further confirmed by quantitative RT-PCR analysis (**Fig. 4G**). These results indicate that the β -cat expression aligns with the VE-CAD expression trend and demonstrate that within HP hydrogels,

heightened EC contractility triggers the activation of focal adhesion kinase, resulting in dissociation of β -cat from VE-CAD mediated adherens junctions, subsequently disrupting intercellular junctions.”

Changes were made: Manuscript, Page 50

Figure 4. Experimental and mathematical simulation show that hydrogel plasticity mediates cell contractility and adherens junctions.

(A) Representative immunofluorescent images of β -cat of HUVECs encapsulated in HP, MP and LP hydrogels. (β -cat in green, nuclei in blue). Scale bar: 25 μ m. (B) Quantification of the normalized intensity of β -cat.

Figure 4. Experimental and mathematical simulation show that hydrogel plasticity mediates cell contractility and adherens junctions.

(C, D) Protein expression levels of β -cat determined by Western blotting of ECs encapsulated in different plastic hydrogels.

Q8. In Figure 3, B and C should be labelled consistently; they currently flipped in the panel.

A8: We thank the reviewer to point out this error and have corrected it in the revised manuscript.

Q9. *Page 3 line 75 remodeling should read remodelling; Page 3 line 76 challenging should read challenging. I encourage a careful read of the manuscript as there are many spelling and grammatical errors throughout.*

A9: We thank the reviewer for the comments. We have revised the typo of “remodeling/challenging” and revised all the grammatical and spelling errors throughout the manuscript.

Response to Reviewer #2

Q1. What plasticity means in the context of the presented work should be briefly addressed in the abstract, if space is available, for the reader more fully appreciate the points being made there.

A1: We appreciate the reviewer for the insightful suggestion. In response, we have included a concise description of mechanical plasticity in the abstract of our manuscript.

Changes were made: Manuscript, Page 2

“The mechanical plasticity of matrix, defined as its ability to permanently deform by external traction, is pivotal in modulating cell behaviors. This is primarily facilitated through enhanced cell-matrix interactions driven by plastic remodeling. Nevertheless, the implications of matrix plasticity on cell-to-cell interactions during EC outgrowth, along with the molecular pathways involved, remain elusive.”

Q2. The examination of the plasticity of tissues at the start of the Results (Figure S1) is interesting. It would be worth considering including these data in the main figure if possible. Additionally, how these data compare to prior work in the literature examining the moduli of tissues (e.g., the work of Shelly Peyton, <https://doi.org/10.1371/journal.pone.0204765>) should be addressed in the Results/Discussion.

A2: We thank the reviewer for pointing these out. Following the suggestions, we have moved the schematics of the creep and recovery tests from fig. S1 to Fig. 1 for better clarity and relevance.

To further address the query about tissue modulus versus tissue plasticity, we performed a comprehensive comparison with established literature (including the work of Shelly Peyton *Plos One* 13, e0204765, (2018), *Nat. Rev. Bioeng.* 1, 60, (2023) and *Prog. Mater. Sci.* 120, 100738 (2021)). Considering the difference in biological species, testing methods or instruments from different papers, it is difficult to carry out the specific comparison of their numerical values. However, we still found the trend that, in general, tissues with high modulus usually exhibit low plasticity while tissues with low modulus are more susceptible to permanent deformation. Moreover, the relationship between modulus and plasticity is not simply linear, which are often influenced by a combination

of factors, including the ECM composition, structure, fiber arrangement, and intercellular interactions of the tissues. Following the reviewer's suggestion, we have added the following paragraph to describe the relationship between plasticity and modulus of tissues in our article.

Changes were made: Manuscript, Page 6

“In addition, we found that tissues boasting a high modulus tend to have reduced plasticity, while softer tissues display a higher propensity to permanent deformation after removing the force. However, the relationship between modulus and plasticity is not merely linear, which is also modulated by various factors such as ECM composition, structural design, fiber orientation, and the nature of tissue intercellular interactions.”

Q3. Collagen should be defined as Collagen I on its first mention in the Results. Additionally, how similar or different the presented materials are from those previously reported in the literature should be noted in the Results and further discussed in the Discussion if needed with relevant references – references to published protocols for building block synthesis are noted in the Methods, but what aspects of the materials formulations are established vs. new needs to be clarified. This information is needed for understanding the potential innovations and impact of the presented work in the context of the literature (e.g., are the materials new, is the application of them new, or are both the materials and application new).

A3: We thank the reviewer for the constructive comments. We have substituted “Collagen” with “Collagen I (Col I)” in the Results section. Furthermore, to clarify the novelty of our materials and their applications, we have made relevant modifications and additions in the Results, Discussion, and Method sections.

In Results section:

Changes were made: Manuscript, Page 7

“Host-guest interaction is a bioorthogonal non-covalent reaction that can undergo rapid dynamic exchange under physiological conditions. Considering their more predictable and reproducible properties based on the precisely defined stoichiometry (1 host + 1 guest complex at each crosslink)^[34], several previous studies have developed host-guest HA hydrogel networks with different dynamics but similar stiffness by mixing various pairs of host-guest complexation (with different binding kinetics)³⁵, however, their

adjustable range of dynamic features tuned by this approach are narrow since both of the complexations are consisted of physical host-guest crosslinks. To address this, a purely elastic network constructed by fully covalent crosslinks was used as static control network in this study. Specifically, we employed ...”

Changes were made: Manuscript, Page 8

“Another important feature in natural ECM of ECs is the fibrillar architecture, which is critical for 3D vasculogenesis and angiogenesis^{12,38}. However, the previously reported dynamic tunable HA hydrogel crosslinked by host-guest interactions only forms nanoporous networks and lack microscale fibrillarity. In this study, we introduced the collagen I (Col I) interpenetrated into the above HA networks to mimic the natural ECM of ECs. The concentration of Col was fixed at 0.2 wt% and the concentration of the HA in every Col-HA hydrogels was kept identical at 2.0 wt% throughout the following experiments.”

In Discussion section:

Changes were made: Manuscript, Page 19

“To understand the role of matrix plasticity in EC outgrowth, a tailored hydrogel system is required that can decouple mechanical plasticity from other physical properties (e.g., stiffness), and also allows the physiological process of vasculature formation. **However, existing hydrogel systems do not support both a multicellular tissue formation and an independently tunable mechanical plasticity.** Although plenty of dynamic HA hydrogels crosslinked by host-guest interactions have been developed for tracking cell fates⁷⁶, their narrow tunable ranges and lack of fibrillar structures limit their applications in studying EC behaviors of vasculogenesis and angiogenesis. In our work, we established a hydrogel system capable of tuning mechanical plasticity, independent of stiffness, by using Col-HA hydrogel systems (HP, MP, and LP hydrogels), with which we unraveled the mechanical plasticity-mediated responses of ECs during vascular vasculogenesis and angiogenesis.”

In Method section:

Changes were made: Manuscript, Page 21-22

“The cyclodextrin-modified host macromolecule (HA-CD) and the adamantane-modified guest macromolecule (AD-HA) were prepared in the same way as previously reported in the literature^[70] **without additional modifications**. Briefly, HA-AD was ...”

“The methacrylated HA (HA-MA) was prepared by the reaction of HA with methacrylic anhydride in a similar way to that reported previously in the literature (**fig. S1B, C**)^[71] **without additional modifications**.”

“**The specific method of mixing the elastic and plastic systems and interspersing Col I in the above network is as follows:** to prepare HP hydrogels, the stock solutions of HA-CD, HA-AD and Col I were uniformly mixed in a stoichiometric ratio of 1:1 for AD and CD. The self-assembly of the host-guest group can be completed in seconds, and the hydrogel needs to be placed at 37°C and incubated for 30 minutes to trigger the polymerization of **Col I**. MP hydrogels were prepared by uniformly mixing stock solutions of HA-CD, HA-AD, Col I, HA-MA with high modification and photoinitiator LAP solutions. Specifically, HA-CD stock solution (3.6 wt% in PBS, pH 7.4) was first mixed with Col I system (including 1X PBS, 0.2 mol NaOH, ECGM, 1 wt% Col I stock solution), HA-MA stock solution (2 wt% in PBS, pH 7.4) and LAP stock solution (5 wt% in PBS, pH 7.4). Then HA-AD stock (2.4 wt% in PBS, pH 7.4) was mixed with the above mixture by a luer lock. The hydrogel was first allowed to polymerize at 37°C for 30 minutes, and then irradiated with blue light (405 nm) for 8 s to trigger covalent crosslinking. LP hydrogels were prepared by uniformly mixing HA-MA solution with low modification, LAP stock solution and Col I solution. The mixture was placed in the 37°C incubator for 30 min, and then irradiated under blue light (405 nm) for 6 s. The Col I was interpenetrated to the above HA networks at a fixed concentration of 0.2 wt% and the concentration of the HA in every sample was kept identical and constant at 2.0 wt% throughout the following experiments. In addition, the photoinitiator content in both LP and MP hydrogels was 0.05 wt%.”

Q4. Use of visible light is noted “rather than traditional UV, avoiding potential cytotoxicity of the reaction.” This point needs a reference or further explanation as there is ample work that supports how low doses of long wavelength UV light are not cytotoxic for many cell types including with the photoinitiator LAP (e.g., the work of Kristi Anseth, <https://www.ncbi.nlm.nih.gov/pmc/articles/PMC2896013/>).

A4: We thank the reviewer for the comments. We agree with the reviewer that some studies have shown the safety of the UV light to cells while others still demonstrated the cytotoxicity of UV which highly depends on the UV intensity, duration and specific wavelength (refer to *Dent. Mater.* 34, 389-399 (2018) and *Biomaterials.* 30, 344-353 (2009)). As a result, compared to UV light, 405 nm visible light has a higher initiation efficiency as well as better cytocompatibility. To clarify this, we have added the related reference and weaken the explanation as follows.

Changes were made: Manuscript Page 7

“The lithium phenyl (2,4,6-trimethylbenzoyl) phosphinate (LAP) as a photoinitiator that can initiate the polymerization under blue light (400-500 nm) with higher initiation efficiency and lower cytotoxicity compared to UV^{36,37}”

Q5. 72 hours is utilized for evaluating HUVEC response in “Hydrogel plasticity promotes 3D vascular morphogenesis....” Why this timepoint was selected needs to be explained / justified.

A5: We thank the reviewer for the comments. Firstly, we would like to point out that we tracked and compared four timepoints of 6 h, 24 h, 48 h and maximum 72 h to evaluate the HUVECs responses instead of single timepoint of 72 h (see **Fig. 2C-G**), since 3D vascular morphogenesis of ECs usually initiates from vacuoles formation at 6-8 h and proceeds with sprouting and branching at 12-24 h to vascular network formation at 48-72 h. In addition, at the timepoint of 72 h from **Fig. 2A**, we have already observed that the comprehensive vascular networks formed in both HP and MP hydrogels, whose networks may collapse or shrink with prolonged culture over this timepoint, leading to the risk of inaccurate statistical analysis of the morphogenesis. To clarify this, we have added the data of **fig. S3** and explanations in the manuscript as follows.

Changes were made: Manuscript. Page 10

“To investigate the role of matrix plasticity on the kinetics of ECs morphogenesis, we encapsulated and tracked human umbilical vein endothelial cells (HUVECs) in the HP,

MP and LP hydrogels at the timepoints of 6 h, 24 h, 48 h and 72 h, respectively (Fig. 2A,B). The following sprouting, branching and complex vasculature formed with featured lumens were sequentially tracked at each timepoints, and the 72 h is selected as the maximum timepoints since the comprehensive vascular networks have been observed in HP hydrogels, which collapse or shrink with further prolonged culture over this timepoint (Fig. S3).”

Changes were made: Supplementary information, Page 5

Fig. S3. Characterization of vascular morphology of HP hydrogels after 72 h.

(A) Representative confocal immunofluorescent images show morphology changes of encapsulated ECs in HP at 72 h, 84 h, and 96 h in incubation (F-actin in red and nuclei in blue). Scale bars: 25 μm . (B-D) Quantitative analysis of vascular tube formation of 72 h, 84 h and 96 h in HP hydrogels including (B) master tube length, (C) number of master junctions, (D) vessel area. All the data are presented as mean values \pm SEM, $n \geq 3$ biological replicates per group, ‘ns’ indicates no statistical difference, * $P < 0.05$, ** $P < 0.01$ and *** $P < 0.001$.

Q6. What number of biological replicates were used (e.g., number of donors or animals) vs. number of sample replicates should be clarified for both *in vitro* and *in vivo* studies with explanation / justification.

A6: We thank the reviewer for the comments and we would like to clarify this as follows. For *in vitro* analyses, based on literature recommendations, we ensured a minimum of three replicates for each hydrogel variant (HP, MP, and LP). Each replicate accounted for 30 cells (refer to *Sci. Adv.* 9, eade9497 (2023) and *Nat. Commun.* 7, 12210 (2016)) and 8-12 spheroids (refer to *Nat. Commun.* 8,16106 (2017) and *Bioact. mater.* 7, 364-376 (2022)), guaranteeing the reliability of our findings. For *in vivo* evaluations, we implanted a minimum of 5 mice per hydrogel group (HP, MP, and LP) subcutaneously (2 samples per mouse) to minimize the risk of obtaining false-positive or false-negative results (refer to *Adv. Funct. Mater.* 22, 2027-2039 (2012)). To counter potential biases, randomized animal assignments and blinded assessments were conducted. For reader clarity, we have incorporated these details into the Methods section and figure captions.

Q7. *It is noted that “We then quantified the expression of Beta1 integrin through Western blot (WB) analysis from the hydrogel (Fig. 3C) and found that the HP hydrogel networks recruit more integrins and enhance cell-matrix interactions.” What is being shown and observed in the figure needs to be further described here. How “recruit more integrins” is be assessed with the WB, which will show expression of integrins at the protein level but not their spatial localization/recruitment, needs to be clarified or perhaps the statement wording adjusted.*

A7: We thank the reviewer for the comments and we agree that the statement of “recruit more integrins” was misleading. Following the reviewer’s suggestion and to clarify this, we have added further descriptions for **Fig. 3C** and revised the statement as follows.

Changes were made: Manuscript. Page 11

“We then quantified the protein levels of integrin β 1 expressed by ECs encapsulated in the HP, MP and LP hydrogels through Western blot (WB) analysis (Fig. 3C**). The quantitative analysis of the WB results demonstrated that the expression of integrin β 1 are significantly higher in the HP hydrogels compared with that in the MP and LP hydrogels (**Fig. 3D**). These results indicate that high hydrogel plasticity promotes the integrin β 1 expression and thus enhances cell-matrix interactions.”**

Q8. *The manuscript, especially the abstract and part of the introduction, has many typographical errors. A thorough proofread is needed.*

A8: We thank the reviewer for pointing this. We have double-checked throughout the manuscript, including the abstract, introduction, method, results and discussion to correct the spelling and grammar errors.

Reviewers' Comments:

Reviewer #1:

Remarks to the Author:

The authors have made a concerted effort to resolve my concerns with new experiments and revised the text significantly to help the reader understand the many aspects. I am satisfied with the manuscript in its current form for publication in Nature Communications.

Reviewer #2:

Remarks to the Author:

The authors have addressed all of the reviewer's concerns in this nice revision. The work will be of broad interest.

Response to Reviewer #1

Q. The authors have made a concerted effort to resolve my concerns with new experiments and revised the text significantly to help the reader understand the many aspects. I am satisfied with the manuscript in its current form for publication in Nature Communications.

A. We thank the reviewer for the encouraging comment.

Response to Reviewer #2

Q. The authors have addressed all of the reviewer's concerns in this nice revision. The work will be of broad interest.

A: We thank the positive comments and encouragement from the reviewer on our revised manuscript.